# A Novel Cystatin Gene from Sea Cucumber (*Apostichopus japonicus*): Characterization and Comparative Expression with Cathepsin L During Early Stage of Hypoxic Exposure-Induced Autolysis

**DOI:** 10.3390/foods14081404

**Published:** 2025-04-18

**Authors:** Siyu Yao, Rui Zhang, Siyuan Ma, Ting Zhao, Qinhao Liu, Lin Zhu, Chang Liu, Liming Sun, Ming Du

**Affiliations:** SKL of Marine Food Processing & Safety Control, National Engineering Research Center of Seafood, School of Food Science and Technology, Dalian Polytechnic University, Dalian 116034, China; 13596655105@163.com (S.Y.); zhangr9965@163.com (R.Z.); 231720860001090@xy.dlpu.edu.cn (S.M.); ztttin1729@163.com (T.Z.); 16639738863@163.com (Q.L.); duming@dlpu.edu.cn (M.D.)

**Keywords:** sea cucumber (*Apostichopus japonicus*), autolysis, cystatin, cathepsin L, hypoxia

## Abstract

Autolysis in sea cucumber has long been a threat to raw material storage and product processing. The involvement of endogenous cysteine protease in sea cucumber autolysis has been proved extendedly. However, as an essential part of the mechanism of autolysis, the role of its endogenous inhibitor has seldom been reported. To investigate the role of cysteine protease inhibitors in the early stage of hypoxic exposure-induced autolysis, a novel cystatin gene (SjCyt) belonging to the subfamily of cystatin C was cloned from *Apostichopus japonicus* by homology cloning and rapid amplification of cDNA ends. The affinity of SjCyt to cysteine protease (cathepsin L and cathepsin B) was investigated by molecular dynamics simulations. Pertinent metrics, including the root mean square deviation, radius of gyration, Gibbs free energy, binding free energy, and bond-forming frequency, showed that the conformation of SjCyt–SjCL was more stable and confirmed a stronger interaction of SjCyt with cathepsin L than with cathepsin B. Thus, cathepsin L (SjCL) was selected to further study its co-expression with SjCyt over a period of 9 h at an early stage of hypoxic exposure. Quantitative RT-qPCR revealed a ubiquitous transcriptional profile of SjCyt and SjCL in all the tested tissues, with the highest abundance in the dorsal epidermis, tube feet, and coelomocytes. Temporal transcription of them showed an overall up-regulated co-expression in the dorsal epidermis and tube feet. However, up-regulated SjCyt and down-regulated SjCL were observed at the protein level. Further immunofluorescence double labeling also found increased staining of SjCyt and SjCyt–SjCL complexes and decreased SjCL. Additionally, recombinant SjCyt was prepared and demonstrated an evident autolysis-inhibiting effect. The results of this study indicated that the anti-autolytic regulation of SjCyt functions at the very early stage of hypoxic exposure, exerting effects at both the transcriptional and translational levels. The above finding offers new insights into the mechanisms of sea cucumber autolysis.

## 1. Introduction

In the last 20 years, sea cucumber (Echinodermata, Holothuroidea) has raised consumers’ interest globally, especially in Asian countries, due to its special texture, high nutritional value, and health-protecting properties. In coastal areas, many people consume sea cucumber for a continuous period each year to improve their health. *Apostichopus japonicus* has consistently maintained a high market value among all marine products in China. So, the farming scale of sea cucumber has been increasing, and the output was 222,707 tons in 2021, 4.56-fold higher than the total production of all the other countries (https://www.fao.org/fishery/statistics-query/en/aquaculture/aquaculture_quantity, accessed on 27 March 2025). In 2022, the gross output of sea cucumber in China reached up to 248,508 tons.

Due to the high commercial price, the typical appearance and texture are considered critical to evaluate the quality of *Apostichopus japonicus* as a raw material or a ready-to-eat or semi-finished product. However, sea cucumbers are vulnerable to a variety of environmental stressors, including variations in oxygen supply, water temperature, salinity, nutrient availability, ultraviolet radiation, and mechanical disturbances. These factors can threaten the viability of sea cucumbers and trigger autolysis during harvesting, transportation, processing, and preservation. Autolysis in sea cucumbers involves epidermal breaking (skin melting), soluble protein degradation [1], and even collapse of the main structural components, like collagen [2]. Thus, autolysis in sea cucumber will cause direct and heavy losses for industry and business. Although environmental factors might trigger autolysis by different signaling pathways, it is predominantly mediated by endogenous (extracellular and intracellular) proteases, such as cysteine protease, matrix metalloproteases, and serine protease [1,2,3,4]. At the present time, almost all the studies on autolysis in sea cucumber have mainly focused on endogenous proteases. It is well known that the activity of endogenous proteases is strictly controlled and regulated by their specific and even non-specific endogenous inhibitors. Nevertheless, little attention has been paid to the role of endogenous protease inhibitors in autolysis.

As mentioned above, cysteine protease has been identified as the principal protease participant in sea cucumber autolysis [1,2]. Research indicates that the biosynthesis of cysteine proteases is controlled through multiple mechanisms, including lysosomal compartmentalization, proenzyme activation, and endogenous protein inhibitors [5]. One of the most well-studied inhibitor families is the cystatins, which act as highly specific, reversible inhibitors of cysteine proteases. Cystatins comprise a large superfamily of evolutionarily conserved proteins, each containing at least one structural domain of approximately 100–120 amino acids with characteristic sequence motifs [5]. Cystatins are broadly distributed and functionally diverse across various species. In mammals, the significance of cystatins has been linked to the pathological processes of various conditions involving tissue damage and protein degradation [6]. In aquatic organisms, the role of cystatins has gained much attention due to their involvement in response to environmental stressors and defense against bacterial invasion, such as cystatin A in *Crassostrea gigas* [7]; cystatin B in *Sebastes schlegelii* [8], *Paralichthys olivaceus* [9], and *Hippocampus abdominalis* [10]; and cystatin C in *Paralichthys olivaceus* [11]. But until now, there has been no report on the characterization of, change in, or the effect of cystatin during autolysis in sea cucumber.

Generally, autolysis in sea cucumbers always occurs after capture and during subsequent transportation or processing. Hypoxic exposure might be the first and most critical stress that a sea cucumber encounters after being caught. Therefore, in this study, adult living sea cucumbers were captured, and autolysis experiments were conducted under hypoxic conditions, without additional UV stimulation, as was utilized in other studies [1,2]. A cystatin-like (SjCyt) gene in sea cucumber was cloned and characterized. The comparative interaction of SjCyt with cathepsin L and cathepsin B was studied by molecular dynamics simulations. And the expression of SjCyt and cathepsin L was studied during the early stage of hypoxic exposure-induced autolysis at the transcriptional and translational levels. The effect of SjCyt on autolysis is discussed.

## 2. Materials and Methods

### 2.1. Experimental Animal and Hypoxia Exposure Experiment

Matured sea cucumbers (*Apostichopus japonicus*), 15–20 cm long and 100 ± 10 g in weight, were obtained from Bangchuidao Group Co. Ltd., in Dalian, China. Five living sea cucumbers were randomly selected. Different tissues, including the dorsal epidermis, tube feet, intestine, respiratory tree, longitudinal muscle, and coelomocytes, were collected from the sea cucumbers. All samples were preserved in RNAstore (TianGen Biotech, Beijing, China) at −80 °C for RNA extraction. Twenty-one randomly selected living sea cucumbers were taken out of seawater and put on a clean stainless-steel tray at ambient temperature (close to that of the marine water). Samples were collected from each group at distinct time intervals (0, 0.5, 1, 3, 5, 7, and 9 h). The dorsal epidermis and tube feet of the sea cucumbers were collected at each time point and stored in RNAstore until RNA extraction. All experiments were conducted in three independent replicates.

### 2.2. RNA Extraction and cDNA Synthesis for Cloning of SjCyt

Total RNA was extracted from tissues of the sea cucumbers using the RNAprep Pure Tissue Kit (TianGen Biotech, China) according to the manufacturer’s instructions. RNase-free DNase I was added to remove genomic DNA. First-strand cDNA was synthesized from total RNA using the SuperRT cDNA Kit (CW Biotech, Beijing, China), where each reaction system contained 2 μg of RNA. Reactions were incubated at 42 °C for 40 min and then at 85 °C for 5 min. The integrity of the RNA was verified by agarose gel electrophoresis, in which the 18S and 28S ribosomal RNA bands had good integrity. All cDNA samples were preserved at −20 °C. According to the homologous sequences of cystatin from Genbank, specific primers were designed. Using the synthesized cDNA, a partial sequence of SjCyt was amplified by PCR. To obtain the full length of SjCyt, 3′–RACE was performed using the 3′–Full RACE Core Set with Prime Script RTase (Takara, Kyoto, Japan), and the primers are listed in Table 1. The PCR product was cloned into a PMD19–T vector (Takara, Japan) and evaluated by bi-directional sequencing.

### 2.3. Bioinformatic and Phylogenetic Analyses

The full-length SjCyt cDNA was analyzed using the BLASTX (non-redundant protein) program developed by NCBI (http://blast.ncbi.nlm.nih.gov/blast, accessed on 18 May 2014). The deduced amino acid sequence of the SjCyt cDNA was analyzed using BIOEDIT software. The molecular mass was calculated and the theoretical isoelectric point was predicted by the computer pI/Mw program (http://web.expasy.org/compute_pi/, accessed on 25 May 2014). Multiple sequence alignments were generated by Clustal X. A phylogenic tree was constructed using the Neighbor–Joining method in the MEGA 11 program, and the reliability of the branching was tested by bootstrap resampling with 1000 pseudo-replicates. The amino acid sequences used in the construction of the phylogenetic tree are shown in the Appendix A. Functional domain prediction of SjCyt proteins was performed using the Simple Modular Architecture Research Tool (SMART: http://smart.embl-heidelberg.de, accessed on 25 May 2014). A computer simulation model was generated using the I—TASSER online service. The structural templates used were obtained from the Cooperative Research Society for Structural Biology Protein Structure Database (RCSB), and the top ten templates were all members of the cystatin superfamily with normalized Z-Scores greater than 1 for the threading alignments.

### 2.4. Molecular Dynamics Simulations and Free Energy Calculation

The initial structures of the SjCyt–cathepsin L (Uniprot: A0A0N7CR66) and SjCyt–cathepsin B (Uniprot: A0A1S5RQP9) complexes were optimized using AlphaFold3. Molecular dynamics (MD) simulations were conducted with the GROMACS software (2023.2), employing the Amber14SB force field and TIP3P water model [12,13]. Energy minimization was performed by the method of the steepest descent algorithm [14] and stopped when energy reached 1000.0 kJ/mol/nm. The complex equilibration was conducted in the NVT and NPT ensembles at a constant temperature and pressure of 298.15 K and 1 bar, respectively, using the v–rescale thermostat and the Parrinello–Rahman barostat method. The distance of the protein from the box margin was 1.0 nm. Na^+^ and CL^−^ were added to the model to a final concentration of 0.15 M to neutralize the system’s electric charge. MD production was performed for a total period of 100 ns under the above conditions, with an equilibration time of 100 ps and a time step of 2 fs [15,16]. According to the root mean square deviation (RMSD) and radius gyration (Rg) results, the simulated trajectories were edited and converted by the gmx trjconv tool, and a three-dimentional Gibbs free energy landscape was plotted by DuIvyTools v0.5.0. The molecular mechanics Poisson–Boltzmann surface area method (MM–PBSA) was used to analyze the last 10 ns of equilibrium, and the total binding free energy (ΔE_T_) between SjCyt and cathepsin L or cathepsin B was calculated according to the equation written below [17]. Free binding energies were calculated using the gromacs package.ΔE_T_ = ΔE_vdW_ + ΔE_ele_ + ΔE_pol_ + ΔE_non−pol_


The terms ΔE_vdW_ and ΔE_ele_ represent the van der Waals and electrostatic interaction energies in the gas phase, respectively. ΔE_pol_ and ΔE_non−pol_ are the polar and non-polar solvation energies, respectively. ΔE_T_ is the total binding energy.

Furthermore, in the last 10 ns of the MD simulations, the binding states of 50 pairs of amino acids from the SjCyt–SjCL and SjCyt–SjCB complexes were recorded, and a heat map was plotted using GraphPad prism 9.5.

### 2.5. Real-Time Quantitative Polymerase Chain Reaction (RT-qPCR)

Total RNA was extracted as described before. The first-strand cDNA was synthesized using the PrimeScript^®^ RT reagent Kit with the cDNA Eraser (Perfect Real Time, Takara, Japan). The expression of SjCyt mRNA was determined by RT-qPCR (ABI, Foster City, CA, USA) using SYBR^®^ Premix Ex TaqTM (Tli RNaseH Plus, Takara, Japan) in a total volume of 20 µL with 10 µL of 2 × SYBR^®^ Premix Ex TaqTM (Tli RNaseH Plus, Takara, Japan), 0.4 μL of ROX Reference Dye II (50×), 0.4 μL of each primer (10 μmol/L), and 2 μL of cDNA template. The RT-qPCR program was 95 °C for 30 s, followed by 40 cycles of 95 °C for 5 s and 60 °C for 34 s. Reactions with a primer pair for the cytochrome gene were utilized as the internal control. The specific primers of SjCyt and SjCL for RT-qPCR are listed in Table 1. The amplification efficiencies of the primers were greater than 90% (Cytb: 95.6%; SjCyt: 103.0%; SjCL: 101.9%). The expected PCR product was in the range of 200–300 bp. All reactions were conducted in triplicate. Transcriptional expression was analyzed by the 2^−ΔΔCt^ method.

### 2.6. Western Blotting

Total protein was extracted from the dorsal epidermis and tube feet by Tris-HCl and subjected to SDS-PAGE (10% separating gel and 5% stacking gel). The protein bands in the gel were transferred to a PVDF membrane (Merck KGaA, Darmstadt, Germany) using a semi-dry transfer apparatus (JunYi, Beijing, China). The membrane was then blocked at 25 °C for 2 h (Beyotime, Shanghai, China). Primary antibodies were diluted 1:800 (anti-cystatin C rabbit pAb; ABclonal, Wuhan, China), 1:1000 (anti-cathepsin L Rabbit pAb; Servicebio, Wuhan, China), and 1:10,000 (β-Actin Rabbit mAb; ABclonal) with diluent (Beyotime) and incubated with the membrane overnight at 4 °C. Excess primary antibodies were then washed off using TBST solution (Beyotime), 3 times, each for 10 min. And the PVDF membrane was incubated with secondary antibody (1:5000, HRP-conjugated goat anti-rabbit IgG; ABclonal) for 2 h at 25 °C. After incubation, the membrane was washed as described above. The PVDF was imaged using BeyoECL Plus under a gel system (Bio-rad, Hercules, CA, USA) exposed for 3 s and analyzed quantitatively using software ImageJ (Java 1.8.0, 32-bit).

### 2.7. Double-Label Immunofluorescence Assays

Sections were deparaffinized in an environment-friendly de-waxing solution (3 × 10 min; Servicebio) and hydrated in anhydrous ethanol (3 × 5 min). Afterward, antigen repair was conducted by immersion of the sections in EDTA (1 mM, pH 8.0) and heating in a microwave oven for a total of 20 min, ceasing twice. After washing in PBS (pH 7.4, 3 × 5 min), endogenous peroxidase in the sections was blocked in 0.3% *v*/*v* hydrogen peroxide solution for 25 min at room temperature (RT). After washing with PBS, the sections were then blocked with 3% BSA for 30 min. Then, primary antibody (anti-cathepsin L rabbit pAb, 1:1000) was applied and incubated for 12 h at 4 °C in a humid chamber. After thorough washing with PBS (3 × 5 min), the secondary antibody (HRP-labeled goat anti-rabbit IgG, 1:3000) was applied and incubated for 1 h at RT. The sections were washed and slightly dried, and a drop of fluorescent dye (iF555-Tyramide) was added, after which they were incubated for 10 min at RT away from light. And then the sections were washed in TBST (3 × 5 min). The other antibody was labeled in the same way (primary antibody: anti-cystatin C rabbit pAb, 1:3000; fluorescent dye: iF488-Tyramide). Cell nuclei were further stained using DAPI for 10 min, followed by washing and slight drying of the sections. Thereafter, a tissue auto-fluorescence quencher was applied to the sections, and they were incubated for 5 min. After rinsing them with running water for 10 min, anti-fade mounting medium was applied, and the sections were closed with micro-cover glasses. Samples without incubation with the primary antibody were used as negative controls. Immunofluorescence images were obtained with a fluorescence microscope (Nikon Eclipse C1, Tokyo, Japan). For image acquisition, excitation (Ex) and emission (Em) wavelengths were used as follows: DAPI (Ex359 nm, Em457 nm), iF488 (Ex491 nm, Em 516 nm), and iF555 (Ex 557 nm, Em 570 nm). The number of fluorescent dots was counted using the ImageJ software and normalized by the number of nuclei.

### 2.8. Expression and Purification of Recombinant SjCyt (rSjCyt)

The coding sequence of SjCyt was ligated into the pET21b (+) vector and transformed into BL21 (DE3) cells by the heat-shock method. The pET21b–SjCyt–BL21 was incubated with 0.5 mM Isopropyl β-D- thiogalactopyranoside (IPTG), at 37 °C, 220 rpm, for 4 h, to enable SjCyt expression. Expressed products were verified by SDS–PAGE. For protein extraction, bacterial cells were lysed in buffer A (50 mmol/L Tris–HCl, 100 mmol/L NaCl, 1 mmol/L EDTA, pH8.0) by ultrasonic treatment at 400 W of power, with each pulse lasting 4 s with an interval of 8 s, for a total duration of 20 min (Xinzhi Biotech, Ningbo, Zhejiang, China). The lysate was then centrifuged at 15,000× *g* for 15 min at 4 °C. The precipitate was dissolved in buffer B (100 mmol/L Tris–HCl, 8 mol/L urea, pH8.0) for 30 min at RT. After centrifugation, the supernatant was dialyzed against 20 mM PBS (pH7.4) overnight. The protein sample was loaded into a pre-equilibrated Ni–IDA 6FF column (Shenggong Biotech, Shanghai, China) using a low-pressure chromatography system at a flow rate of 1 mL/min. The column was washed, and proteins were eluted using specific buffers (20 mmol/L Tris–HCl, 250 mmol/L imidazole, 0.15 mol/L NaCl, pH 8.0), followed by overnight dialysis in PBS (pH 7.4).

### 2.9. Effect of Recombinant SjCyt on Autolysis in Sea Cucumber by SDS-PAGE

The dorsal parts of live sea cucumbers were homogenized at 0 °C (dipped in ice water). The mince was divided into five tubes, with 2 g in each. The samples were incubated in the presence or absence of recombinant SjCyt in 50 mmol/L Tris-HCl solution (pH 7.5, 1:1, *v*:*v*) at 25 °C for 0, 9, and 18 h. To verify the protein degradation profile and the effect of recombinant SjCyt on autolysis, soluble protein was extracted by adding extraction buffer (2×, 2% SDS, 100 mM Tris-HCl, 20% glycerol, 8 M urea, 4% β-ME, 0.02% BPB, pH 6.8) into the tube to a final volume of 4 mL. After shaking for 1 h, the mixture was heated at 100 °C for 10 min and centrifuged at 15,000× *g* for 15 min. SDS-PAGE was performed using a 5% stacking gel and 10% separating gel.

### 2.10. Statistical Analyses

The statistical analyses of the data were performed using one-way ANOVA in GraphPad Prism 9.5 (Graphpad software Inc., San Diego, CA, USA). Normal distribution and variance chi-square tests were performed using SPASS to ensure that the data met ANOVA assumptions. The qPCR data were log2-transformed. The data were represented as means ± standard deviations, and *p* < 0.05 was considered statistically significant after correction.

## 3. Results

### 3.1. Identification and Bioinformatic Analysis of SjCyt

As shown in Figure 1, a full-length 662 bp cDNA sequence was obtained by splicing the 3′RACE product with SjCyt-specific gene fragments. Analysis revealed that the complete cDNA of SjCyt contained a 5′ untranslated region of 22 bp and an open reading frame 348 bp in length, encoding 115 amino acids. The identified sequence was submitted to the NCBI GenBank database under accession no. KR872411. SMART analysis revealed that the deduced amino acid sequence of SjCyt comprises a cystatin-like domain (from Ser3 to Ser113). No signal peptide was identified using the SignalP program. The deduced protein was predicted to have a molecular weight (MW) of 12.9 kDa, with a theoretical isoelectric point of 8.65.

The amino acid sequence of SjCyt was compared with that of cystatins from other species (Table 2). As shown in Figure 2A, multiple sequence alignment revealed highly conserved domains characteristic of the cystatins, including the N-terminal glycine (Gly, G) residue, the QXVXG (Q, Gln; V, Val; X, any amino acid) motif, and the C-terminal proline-tryptophan (PW) motif. In SjCyt, QXVXG motifs were identified as QVVSG. And the typical PW motif was replaced by a leucine-tryptophan (LW) pair near the C terminus. Four conserved cysteine residues, which might be involved in forming two intra-chain disulfide bonds, were found at the 101, 117, 131, and 151 amino acid sites. These conserved residues are involved in forming a hydrophobic, tripartite, wedged-shaped edge to penetrate the active site of cysteine protease to form a tight inhibition [18].

The 3D structure of SjCyt (Figure 2B) consists of a five-turn α-helix core surrounded by an eight-stranded antiparallel β-sheet. The highly conserved pentapeptide motif (QxVxG) is located in the first binding loop (between the third and fourth β-sheets at the N terminal), while the LW loop resides in the second binding loop (between the sixth and seventh β-sheets). To assess the phylogenetic relationships of SjCyt with cysteine protease inhibitors from other species, a Neighbor-Joining phylogenetic tree was constructed basing on the deduced amino acid sequences of 22 previously annotated cystatins and kininogens. As shown in Figure 2C, the phylogenetic tree divides cysteine protease inhibitors into three distinct groups. SjCyt clusters closely with cystatin C-like proteins from *Patiria miniata* (Echinodermata), belonging to the cystatin C subfamily.

Based on molecular weight, sequence homology, and the positions of intra-chain disulfide bonds, the cysteine protease inhibitor superfamily is divided into at least three subfamilies, including stefins, cystatins, and kininogens. The stefin subfamily lacks disulfide bonds and carbohydrates. The cystatin subfamily is composed of about 115 amino acid residues with a MW of about 13 kDa. And members of this subfamily are characterized by having two disulfide bonds towards the C terminus. The kininogen subfamily typically consist of glycosylated proteins with a MW from 60 to 120 kDa [19]. From the results in Figure 2, SjCyt belongs to the subfamily of cystatin C.

### 3.2. Molecular Dynamics Simulations

The complexes of SjCyt–SjCB and SjCyt–SjCL were evaluated for their binding stability by MD simulations running for 100 ns in natural conditions (25 °C, 1 bar). Figure 3A shows that the RMSD curve of SjCyt–SjCL came to equilibrium soon after 20 s, and the average RMSD value was 0.167 nm. And the RMSD value of the complex of SjCyt–SjCB (Figure 3B) was stable until 38 s of the simulation, with a higher average value of 0.225 nm. A significant RMSD transition occurred at 82 s. Similarly, the Rg value of SjCyt–SjCL (Figure 3C) remained stable in each dimension during the whole simulations of 100 ns. In contrast, the Rg value of SjCyt–SjCB (Figure 3D) presented a slow fluctuation throughout the simulation process in the X dimension. A clear transition also occurred at about 30 ns in the Y and Z dimensions. But the average Rg value of SjCyt–SjCL was similar to that of SjCyt–SjCB, indicating the similar tightness of the two complexes.

The involvement of cysteine proteases in sea cucumber autolysis has been confirmed, among which the activities of cathepsin B and L (SjCB and SjCL) are the ones of greatest concern [2,20,21,22]. But the effect of SjCyt on them has seldom been reported. Therefore, MD simulations were conducted to compare the stability of the SjCyt–SjCB and SjCyt–SjCL complexes. The RMSD value, which is calculated based on the amide-bond backbone atoms (alpha carbon atoms), is a parameter commonly used to measure the deviation of a protein from its initial conformation at a given time and to examine if the target molecular complex has reached a stable state [23]. A smaller RMSD value means a more stable simulated system. The tightness of a complex conformation could be assessed by the value of Rg, and a smaller Rg value shows a better tightness of the molecular structure [23]. The RMSD and Rg results in Figure 3 indicated that the conformation dynamics of SjCyt–SjCL were more stable than those of SjCyt–SjCB. Therefore, SjCyt seems to have a stronger binding affinity to SjCL than to SjCB.

Moreover, the Gibbs free energy landscape (GFEL) was also introduced to compare the binding stability of the SjCyt–SjCL and SjCyt–SjCB complexes. As shown in Figure 4A,B, the GFEL is a 2D/3D image of the free energy change along with the binding state of SjCyt to SjCL (or SjCB). A single dark-blue energy cluster reflects a stable state, and, conversely, multiple light-red energy clusters represent an unstable state. Inverted mountain-like pits in the GFEL correspond to distinct low-energy conformations, and deeper pits typically indicate greater stability [24]. Upon the formation of SjCyt–SjCL and SjCyt–SjCB complexes, their GFELs both presented single dark-blue clusters, indicating that SjCyt was able to bind to SjCL or SjCB in a state with minimized energy, and thus the formation of the two complexes was spontaneous and exothermic. Comparatively, the blue area in the GFEL of the SjCyt–SjCL complex was darker and lower, consistent with the result shown in Figure 3. Furthermore, the 3D aligned structure of the complex (Figure 4C,D) showed the binding of SjCyt to the groove of the SjCL or SjCB pocket and also revealed that SjCyt–SjCL had a more compact and tight conformation than SjCyt–SjCB.

To quantitatively analyze the binding affinity between SjCyt and SjCL/SjCB, the binding free energy was calculated using the MM–PBSA method basing on trajectory data from MD simulations. The estimated binding free energy of the complex usually includes contributions from the gas phase (ΔE_gas_) and the solvation phase (ΔE_solv_). ΔE_gas_ includes van der Waals and electrostatic potential energy, and ΔE_solv_ includes polar and non-polar energy. Figure 5 shows that the ΔE_gas_ values of SjCyt–SjCL and SjCyt–SjCB were −656 and −505, respectively. Their positive ΔE_solv_ values were 465 and 423, respectively. And their total binding free energy (ΔE_T_) values were calculated to be −192 and −82, respectively.

Among the interactions that contribute to binding free energy, van der Waals interactions are the most significant. In the binding of SjCyt to SjCL/SjCB, the calculated van der Waals energy, electrostatic interaction energy, and non-polar solvent interaction energy are negative, indicating that the formation of the complexes is favored, while the calculated polar solvent interaction energy is positive, hindering the complexes’ formation. Above all, the ΔE_T_ of the two complexes is a negative value, proving the tendency of SjCyt to bind to SjCL/SjCB. Moreover, the ΔE_T_ of the SjCyt–SjCL complex was 2.3 times that of SjCyt–SjCB, suggesting, again, the stronger affinity between SjCyt and SjCL.

Moreover, the binding states of 50 pairs of amino acid residues from each side of the SjCyt–SjCL and SjCyt–SjCB complexes were recorded in the last 10 ns of the simulations, as shown in Figure 6. The overall bond-forming frequency of SjCyt–SjCL was obviously higher than that of SjCyt–SjCB, consistent with the binding free energy results. The areas around the QXVXSG and LW loops are the primary functional regions of SjCyt. Within the above regions, there are at least six amino acid residue sites involved in forming bonds with SjCL. Among the six sites, the bond-forming frequencies of SjCyt–Val51 and SjCL–His280 (in the active site) reached a high value of 0.98, highlighting the critical interaction region between SjCyt and SjCL [21]. In contrast, there were only two sites of SjCyt involved in bonding with SjCB.

### 3.3. Tissue Distribution of SjCyt and SjCL

The mRNA expression of *SjCyt* in tissues of sea cucumbers was detected by quantitative RT-qPCR, standardized against cytochrome b [25] and normalized to the expression level in longitudinal muscle (least expression). As shown in Figure 7A,B, SjCyt and SjCL transcripts were ubiquitously expressed acrose all the tested tissues, with highest levels detectd in the dorsal epidermis, tube feet, and coelomocytes. Lower expression was observed in the respiratory tree and intestine, while the longitudinal muscle exhibited the lowest abundance.

### 3.4. Temporal Expression of SjCyt and SjCL

The dorsum and abdomen are the main edible parts of sea cucumber, and the transcriptional levels of *SjCyt* and *SjCL* in the dorsal epidermis (dorsum) and tube feet (abdomen) are relatively higher than in other tissues (except the coelomocytes). Therefore, the above two parts were used as material to further investigate the transcriptional and translational profiles of SjCyt and SjCL in the early 9 h period of hypoxia exposure. Cytochrome b was used as a standard, and the expression levels at 0 h were normalized. Cytochrome b was used as a standard and normalized to the expression level of 0 h by the ratio method.

The transcriptional profiles were shown in Figure 7C,D. In response to external hypoxia stimuli, an overall periodic fluctuation in the expression of SjCyt and SjCL was recorded throughout the 9 h of observation. In the dorsal epidermis, peak expression of SjCyt and SjCL occurred at 0.5 h and 7 h. In the tube feet, the peak expression occurred at 1 h and 9 h. Obviously, the response in the tube feet was a bit later than that in the dorsal epidermis.

In the dorsal epidermis, a rapid transcriptional response of SjCyt and SjCL, as early as 0.5 h, was up-regulated 3.37- and 3.05-fold, respectively. And the second peak expression of them at 7 h was almost at the same level as the first one. However, in the tube feet, the first response peak of SjCyt and SjCL appeared after 1 h, with 2.28- and 1.97-fold increases, respectively. It is interesting that the second peak, a sudden jump in both SjCyt and SjCL, was measured at 9 h, with 4.95- and 3.98-fold increases, respectively (Figure 7C,D). Notably, in the tube feet and the dorsal epidermis, the SjCyt and SjCL transcriptional profiles were highly similar, suggesting a coordinated up-regulation of them upon hypoxia.

The translational profiles of SjCyt and SjCL are shown in Figure 8. The expression of SjCyt in the dorsal epidermis (Figure 8C) also presented a fluctuating profile, with higher levels at 1 h and 9 h. But SjCyt expression in the tube feet showed a steady increase before 5 h, followed by a drop at 9 h (Figure 8A). Significant depression of SjCL was measured at 1 h in both the dorsal epidermis and tube feet (Figure 8B,D). And then SjCL presented a slow decrease (Figure 8B) or increase (Figure 8D) along with the time of hypoxia exposure.

### 3.5. Distribution of SjCyt and SjCL in the Dorsum

The sound appearance and texture of the dorsum is of the utmost concern for the producer, enterprise, and consumer. Hence, the dorsum was selected to investigate the distribution of SjCyt and SjCL by immunofluorescence double labeling, which could provide their quantitative and location changes in the early stage of hypoxia exposure. Besides lysosomes, studies have demonstrated that active cathepsins are localized in other cellular compartments, such as the nucleus, cytoplasm, and plasma membrane, as well as the extracellular space, and thus that they participate in a wide range of physiological and pathological processes [26]. The catalytically active cathepsin L localized to the nucleus plays a role in the regulation of cell behavior [27,28]. So, the distribution or location of SjCyt and SjCL was also analyzed according to the distance to the nucleus.

As shown in Figure 9, the nuclei were labeled by DAPI with blue fluorescence. SjCyt and SjCL were labeled by iF488-Tyramide (green) and iF555-Tyramide (red), respectively. The relative numbers of labeled SjCyt and SjCL were normalized to the number of nuclear fluorescent dots.

The somatic layer of sea cucumber is full of colloidal matter, mainly collagen and polysaccharides. Consistently, the images of nuclei in Figure 9 show that the cells in the somatic layer are scattered without any specialized arrangement. The green fluorescence of SjCyt dispersed across the image of Figure 9A suggests that it is probably a secreted membrane-permeable protein and plays a role both intra- and extracellularly. Compared with 0 h, a stronger labeling of SjCyt was observed at 1 h (Figure 9B). The merged image analysis also revealed a significant increase in peri-nuclear SjCyt. Furthermore, the distal-nuclear SjCyt fluorescence dots also increased almost 2.0-fold (Figure 9C). According to the image, most of the distal-nuclear SjCyt might be extracellular. In contrast, compared with 0 h, the labeling of SjCL, both peri-nuclear and distal-nuclear, become much less at 1 h (Figure 9C).

Compared with 0 h, the SjCyt–SjCL complex (merged images of SjCyt and SjCL) in the distal-nuclear region also increased apparently. However, the peri-nuclear SjCyt–SjCL complex did not change significantly. It is noteworthy that the dots of the SjCyt–SjCL complex in the distal-nuclear region were far more numerous than those of the peri-nucleus. This meant that most of the SjCyt-binding SjCL might be located in the distal-nuclear area. And the increased number of complex dots further suggested that SjCyt and SjCL might migrate from peri-nuclear lysosomes to the distal-nuclear region within the cell, and potentially even into the extracellular space. Overall, the increased SjCyt and decreased SjCL fluorescence were consistent with the results shown in Figure 8.

### 3.6. Overexpression, Purification, and Activity Validation of Recombinant SjCyt

As shown in Figure 10, recombinant SjCyt (rSjCyt) protein was expressed in *Escherichia coli* harboring the expression vector and coding sequence of SjCyt. SDS-PAGE analysis (Figure 10A) revealed that the rSjCyt protein predominately localized in the precipitate, suggesting that the expressed rSjCyt mainly resided in the form of inclusion bodies. The purified rSjCyt protein was confirmed by SDS-PAGE, and Western blot validation was conducted. After denaturation and renaturation, soluble rSjCyt was obtained from inclusion bodies and then purified by Ni-IDA 6FF affinity chromatography. The purified rSjCyt displayed a molecular weight of approximately 13 kDa (Figure 10B), consistent with the expected size stated in Section 3.1. The effect of rSjCyt on autolysis in the sea cucumber somatic layer is shown in Figure 10C. After 9 h and 18 h of incubation, degradation of the protein band of 200 kDa could be observed by SDS-PAGE (Figure 10C). By adding 12.5 and 25 μg/g of rSjCyt to the mince of the somatic layer, the degradation of band 200 kDa was attenuated dramatically. It is evident that SjCyt is effective in inhibiting the lysosomal cysteine protease during hypoxia-induced autolysis in sea cucumber.

## 4. Discussion

After harvest, hypoxia exposure is one of the direct external stimuli that can induce autolysis in sea cucumber. This autolysis is characterized by the gradual rupture of the dorsal epidermis and the corruption of the body structure. Autolysis is a complex physiological process, during which the protein metabolic profile is altered significantly [29,30]. It is well established that the endogenous proteases have long been regarded as the primary and direct factors responsible for protein degradation during autolysis. However, little attention has been paid to the change and effect of the corresponding endogenous protease inhibitors. Therefore, in the present study, a cystatin C-like gene (SjCyt) was cloned from sea cucumber (*Apostichopus japonicus*). The interaction between this gene and cysteine proteases, as well as their comparative expression levels, were investigated to elucidate its role in the early stage of autolysis.

The interaction between SjCyt and two well-known cysteine proteases in sea cucumber, cathepsin L and cathepsin B, was explored by molecular dynamics simulations. The results showed that SjCyt could form a stable complex with both of these proteases. However, the interaction between SjCyt and cathepsin L (SjCL) was stronger and more stable than that between SjCyt and cathepsin B. Consequently, a subsequent comparative expression study was conducted, focusing on SjCyt and SjCL. Quantitative RT-qPCR revealed a ubiquitous transcriptional profile of SjCyt and SjCL in all the tested tissues. This finding supports the prior knowledge that cystatin C and cathepsin L have a broad distribution across different tissues [11,31,32,33,34]. Moreover, the highest transcriptional abundance of SjCyt and SjCL was found in the dorsal epidermis, tube feet, and coelomocytes.

Sea cucumbers are named for their soft ellipse shape, which resembles a fat hollow cucumber. The thick, gelatinous somatic layer (also referred to as the body wall) constitutes the main edible part of sea cucumbers. The appearance and texture of this layer are the most concerning aspects for consumers. The somatic layer can be divided into the relatively flat abdomen and the slightly elevated dorsum. Numerous tiny tube-shaped feet are distributed along the abdomen of the body. These tube feet mainly serve to anchor the limbless body to the seafloor. Sea cucumbers have no well-developed nerve system; instead, they only have a nerve ring encircling the mouth and five radial nerves radiating into the ellipse somatic layer [35,36]. The coelomocytes in the coelomic fluid of sea cucumbers function as immune cells. They circulate throughout the body to maintain physiological equilibrium [37]. The dorsum, tube feet, and coelomocytes are the main parts through which sea cumbers respond to external and internal stimuli. Thus, it is understandable that higher expression levels of *SjCyt* and *SjCL* were found in these parts. This further suggests that *SjCyt* and *SjCL* might be involved in the stress response of sea cucumber.

Aquatic animals could be exposed to hypoxia during tides and harvest. Considerable transcriptional changes in response to hypoxia have also been observed in aquatic organisms [38,39]. Previous studies have shown that cystatin C [11,32,33,34,35,36,37,38,39,40,41] and cathepsin L [42,43,44] are involved in the rapid responses of both aquatic and terrestrial animals to various stimuli, including lipopolysaccharide (LPS), pathogenic bacteria, and virulent infection. During a 9 h period of hypoxic exposure, the temporal transcription of SjCyt and SjCL exhibited an overall up-regulated co-expression in the dorsal dermis and tube feet. However, at the protein level, SjCyt and SjCL were not co-expressed. Instead, SjCyt was up-regulated, while SjCL was down-regulated. It is well established that the correlation between transcription and translation expression is relatively weak. Research on the correlation between protein and mRNA expression has revealed that about 40% of the variance in protein expression can be attributed to changes at the transcription level, while about 60% is due to other factors, such as post-transcriptional and post-translational regulation, as well as other sources of error [45]. Although a rapid transcriptional up-regulation of SjCyt and SjCL occurred following hypoxia exposure, the key factors influencing autolysis (degradation of extra- and intracellular matrices) depend on their actual, especially the activated, protein levels. As a crucial factor implicated in autolysis, the protein content of SjCL must be strictly regulated for sea cucumbers to repair or regenerate after hypoxia exposure. The significant decrease in SjCL protein content (Figure 8) might result from reduced synthesis and increased activation [25,46]. This might be a reasonable cellular feedback mechanism for sea cucumbers to protect themselves from excessive autolysis. Similarly, a decreased SjCL protein expression was also reported by Dong et al. [29].

On the other hand, the increased synthesis of SjCyt protein further protected the sea cucumbers from inappropriate cellular degradation caused by SjCL and other proteases leaking from lysosomes. Apart from inhibiting the activity of cysteine proteases, cystatin could also reduce the protein amount of target cysteine protease. Wilder et al. treated MDA-MB-231 breast cancer cells with cystatin C and found that the cathepsin L protein level decreased significantly. This might be a feedback mechanism within the proteolytic cathepsin network in response to the inhibitory effect of cystatin [47]. Moreover, abnormal autocatalytic activation of SjCL by glycosaminoglycan in the extracellular matrix of sea cucumbers might be another mechanism leading to the decrease in SjCL protein levels [48].

The interaction between cystatin and cysteine protease is far more than just forming cystatin–cysteine protease complexes with an isomolar ratio. Early studies found that cysteine protease with a blocked active center could still bind to cystatin [19]. Furthermore, cysteine protease was capable of hydrolyzing cystatin [49,50]. The resulting truncated cystatin fragment exhibited significantly reduced affinity for the cysteine protease. Thus, the significant increase in SjCyt expression after hypoxia exposure was a reasonable mechanism for the sea cucumber to modulate the activation of cysteine protease. As is generally accepted, the balance between free cysteine proteases and their complexes with inhibitors is critical for tissue integrity and maintenance of proper function [19]. Therefore, in the future, it will be necessary to uncover the post-harvest disequilibrium between SjCyt and SjCL, as well as other closely related interactions, during autolysis in sea cucumber.

## 5. Conclusions

A novel endogenous inhibitor of cysteine protease, *Apostichopus japonicus* cystatin (SjCyt), was obtained and characterized. SjCyt exhibits ubiquitous tissue expression, with particularly high levels in the somatic layer (dorsal epidermis and tube feet) and coelomocytes. SjCyt could bind preferentially to cathepsin L, forming a more stable complex than with cathepsin B. Hypoxia triggers a highly dynamic transcriptional and translational regulation of SjCyt and SjCL in sea cucumbers. SjCyt exhibits diffuse distribution throughout the dorsal epidermis and demonstrates rapid responsiveness to post-harvest hypoxia, showing significant up-regulation at both transcriptional and translational levels during early hypoxic exposure. In contrast, cathepsin L (SjCL) displays divergent regulation—transcriptional activation but translational suppression—suggesting post-transcriptional control.

The concurrent increase in SjCyt protein levels and SjCyt–SjCL complex formation, coupled with decreased SjCL activity, implies that SjCyt may regulate SjCL through translational inhibition and/or complex stabilization. Functional validation using prokaryotic expressed recombinant SjCyt confirmed its anti-autolytic role, significantly reducing protein degradation in sea cucumber mince.

Thus, the following key conclusions can be drawn: SjCyt acts as an early-response anti-autolysis factor, rapidly induced upon hypoxia. A regulatory imbalance exists between SjCyt (transcriptional and translational up-regulation) and SjCL (translational down-regulation). The SjCyt–SjCL axis represents a novel checkpoint in hypoxia-induced autolysis.

This study provides the first mechanistic insights into cystatin-mediated protease regulation during post-harvest autolysis in *Apostichopus japonicus*, offering potential applications for sea cucumber quality control.

## Figures and Tables

**Figure 1 foods-14-01404-f001:**
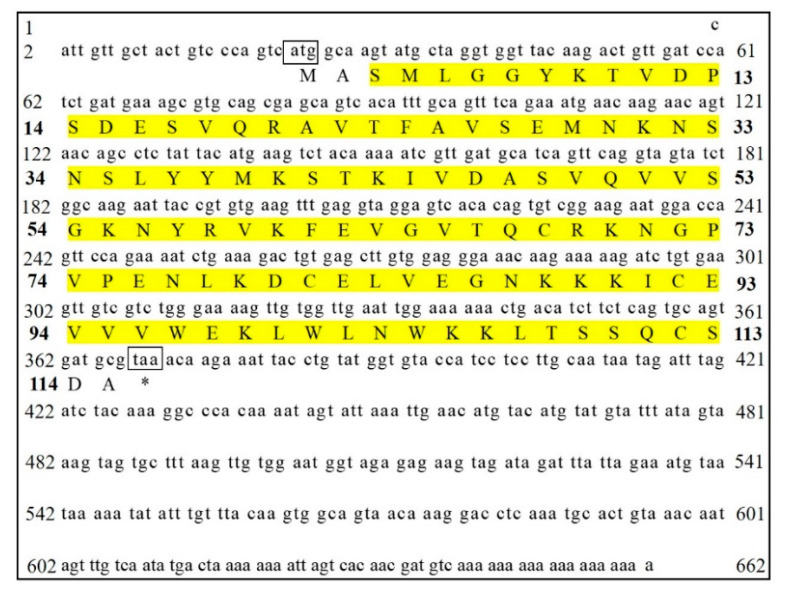
Nucleotide and predicted amino acid sequences of SjCyt. The start codon (atg) and stop codon (taa) are boxed, and the amino acids that constitute the cystatin-like domain are highlighted in yellow.

**Figure 2 foods-14-01404-f002:**
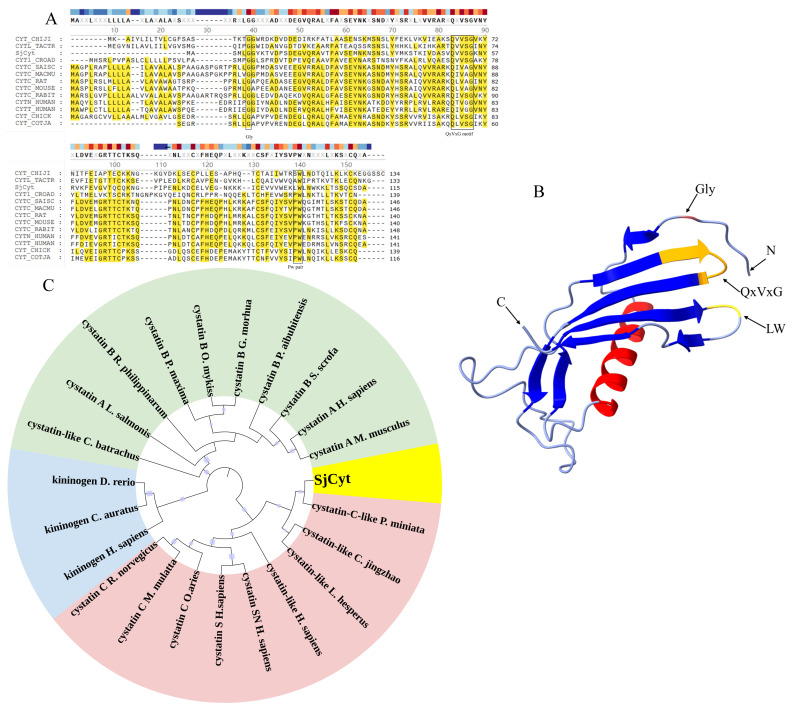
Conservation and homology analysis of SjCyt. (**A**) Multiple sequence alignments of SjCyt with other cystatins. The highly conversed amino acid residues are red and underlined with labels. (**B**) Three-dimensional structure of SjCyt. The α-helix is shown in red; the β-folding is shown in blue; glycine residues are shown in red; the QxVxG loop is shown in orange; the LW loop is shown in yellow. (**C**) Phylogenetic tree of SjCyt with some known species of cysteine protease inhibitors. Green for family 1 (stefins); pink and yellow for family 2 (cystatins); blue for family 3 (kininogens).

**Figure 3 foods-14-01404-f003:**
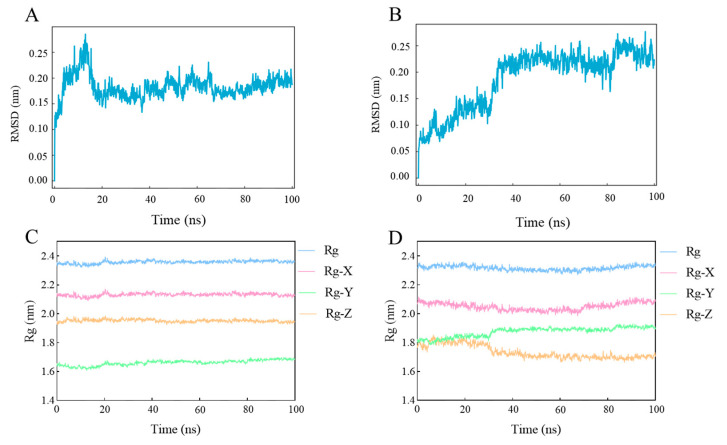
Root mean square deviation (RMSD) and radius of gyration (Rg) of the SjCyt–SjCL and SjCyt–SjCB complexes during MD simulations of 100 ns. (**A**,**C**) RMSD values of SjCyt–SjCL and SjCyt–SjCB, respectively. (**B**,**D**) Rg values of SjCyt–SjCL and SjCyt–SjCB, respectively.

**Figure 4 foods-14-01404-f004:**
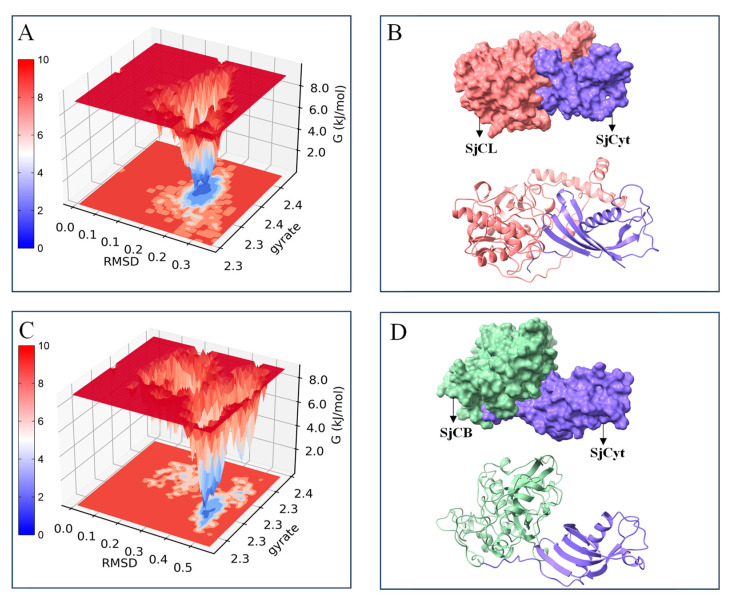
Gibbs free energy landscapes and aligned 3D structures of the complexes in kinetically stabilized phases. (**A**,**B**) SjCyt–SjCL. (**C**,**D**) SjCyt–SjCB.

**Figure 5 foods-14-01404-f005:**
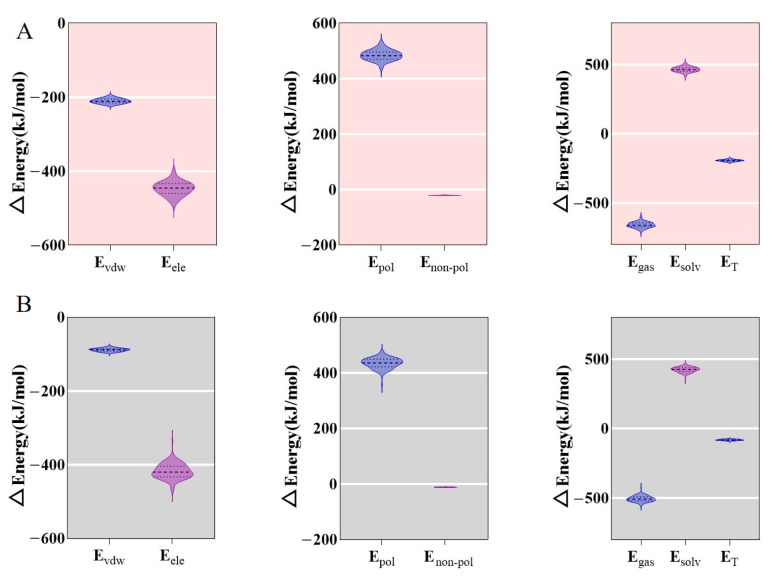
Binding free energy between SjCyt and cathepsin L or cathepsin B. (**A**) SjCyt and SjCL. (**B**) SjCyt and SjCB. E_vdw_, van der Waals energy; E_ele_, electrostatic energy; E_pol_, polarization energy; E_non−pol_, non-polar energy; E_gas_, free energy of gas phase; E_solv_, free energy of solvation phase; E_T_, total binding energy. ΔE_gas_ = ΔE_vdw_ + ΔE_ele_; ΔE_solv_ = ΔE_pol_ + Δ_Enon−pol_; ΔE_T_ = ΔE_vdw_ + ΔE_ele_ + ΔE_pol_ + ΔE_non−pol_.

**Figure 6 foods-14-01404-f006:**
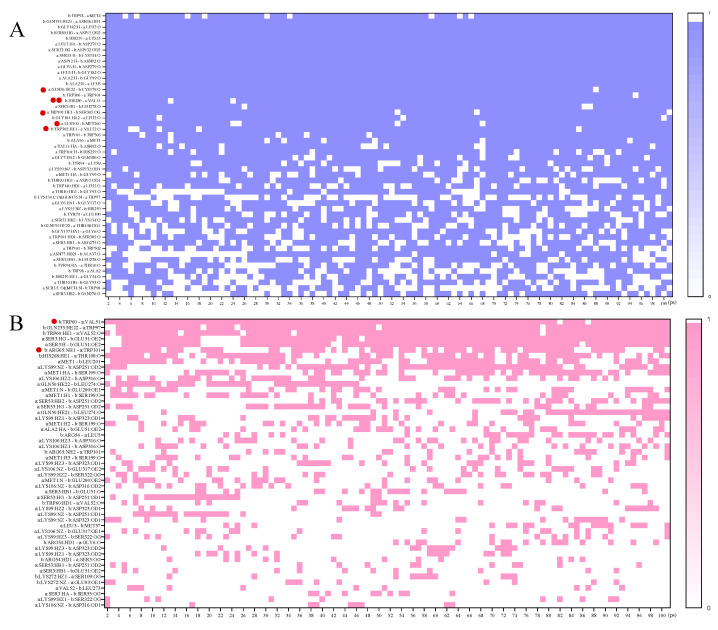
Bond-forming frequencies of the 50 pairs of amino acid residues from SjCyt–SjCL and SjCyt–SjCB in the last 100 ps of simulations. (**A**) SjCyt (marked with a lowercase a) and SjCL (marked with a lowercase b). (**B**) SjCyt (a) and SjCB (b). The amino acid residues in the active region of SjCyt that binds to SjCL or SjCB with a frequency greater than 0.8 are marked by a solid red disc on the left. Two solid red discs indicate a bond formed by the active sites of both SjCyt and SjCL. Blue (or pink) means bonded, white means not bonded.

**Figure 7 foods-14-01404-f007:**
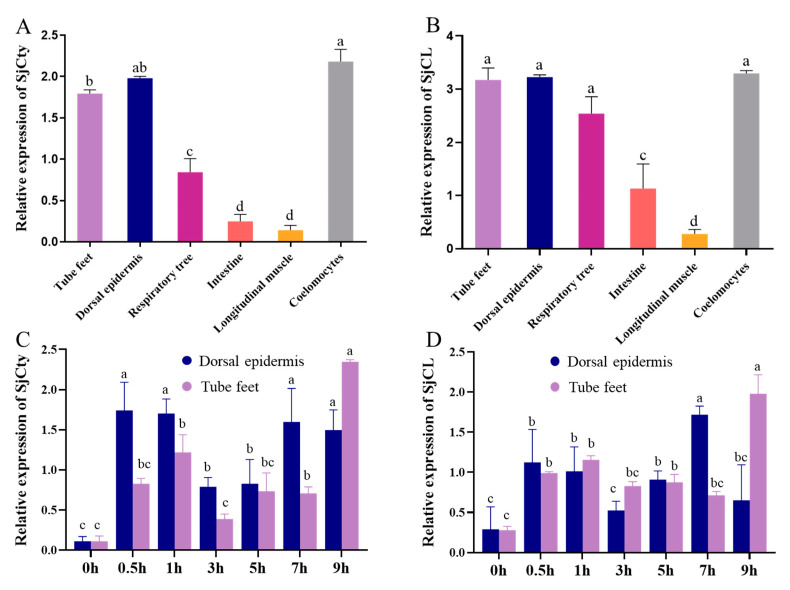
Transcriptional profiles of SjCyt and SjCL. (**A**,**B**) Expression patterns of SjCyt and SjCL in different tissues, respectively. (**C**,**D**) Temporal expression of SjCyt and SjCL in dorsal epidermis and tube feet, respectively. In the same group, different letters represent significant differences (*p* < 0.05).

**Figure 8 foods-14-01404-f008:**
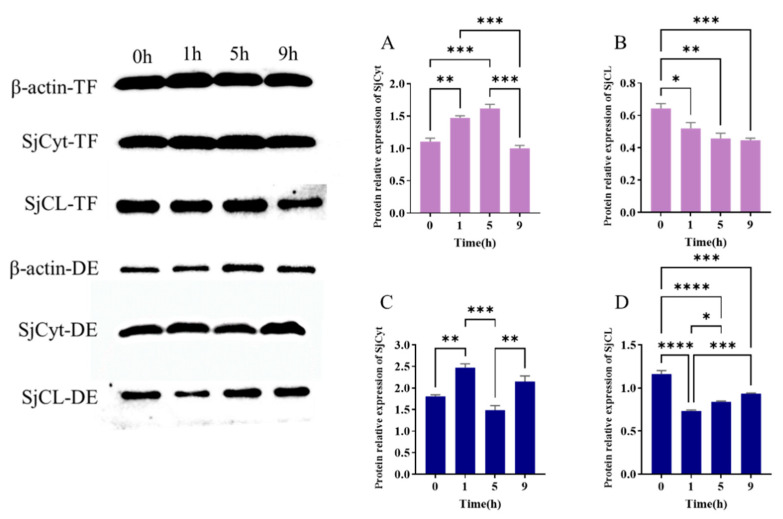
Translational profiles of SjCyt and SjCL at 0, 1, 5, and 9 h of hypoxia exposure. (**A**) SjCyt in tube feet. (**B**) SjCL in tube feet. (**C**) SjCyt in dorsal epidermis. (**D**) SjCL in dorsal epidermis. *: *p* < 0.05; **: *p* < 0.01; ***: *p* < 0.001; ****: *p* < 0.0001.

**Figure 9 foods-14-01404-f009:**
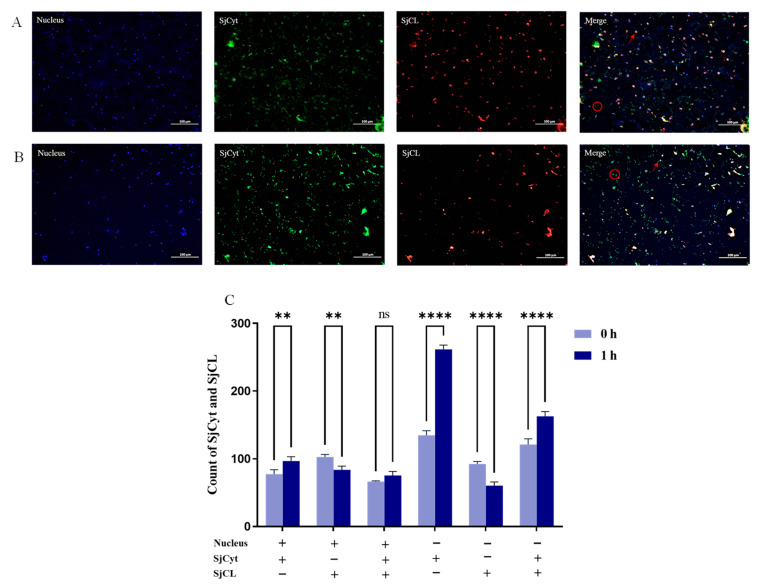
Double-labeling immunofluorescence of SjCyt and SjCL in dorsal epidermis of sea cucumber. (**A**) Hypoxia exposure for 0 h. (**B**) Hypoxia exposure for 1 h. (**C**) Quantitative analysis of the immunofluorescence results. SjCyt is labeled in green, SjCL in red, and nuclei in blue. Arrows indicate peri-nuclei and circles indicate distal nuclei. ns: not statistically significant; **: *p* < 0.01; ****: *p* < 0.0001.

**Figure 10 foods-14-01404-f010:**
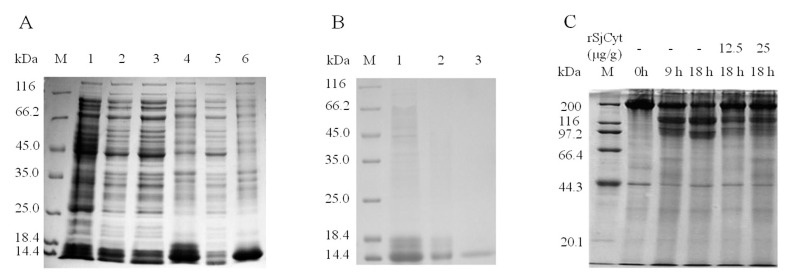
Preparation and autolysis-inhibiting activity of rSjCyt. (**A**) Expression of rSjCyt in *E. coli*. M: marker, 1: not induced by IPTG; 2, 3, 4, 5: induced by IPTG; 3, 5: supernatant of the *E. coli* culture product at 11 °C and 37 °C, respectively; 4, 6: precipitate of the *E. coli* culture product at 11 °C and 37 °C, respectively. (**B**) Isolation and purification of rSjCyt by Ni column. M: marker 1: unpurified protein; 2: flow-through part; 3: purified rSjCyt. (**C**) Effects of rSjCyt addition on the degradation of sea cucumber somatic layer protein.

**Table 1 foods-14-01404-t001:** List of primer sequences used in this study.

Name	GenBank	Primer Sequence (5′–3′)	Purpose	Tm (°C)
SjCyt–F	/	TGCTACTGTCCCAGTCATGG	Homologous cloning	/
SjCyt–R		TAATACTATTTTGTGGGCCTTTG		
SjCyt–O	/	GTTGTCGTCTGGGAAAAGTTGTG	3′RACE	/
SjCyt–I		GATGCGTAAACAAGAAATTACCTGT		
SjCyt–RTF	KR872411	GTGAAGTTGTCGTCTGGGAAAAG	Real-time PCR	79
SjCyt–RTR		AGGAGGATGGTACACCATACAGG		
SjCL–RTF	EU143709	GCCAGCCACGAGTCTTTCCAA	Real-time PCR	85
SjCL–RTR		CGATCCAGTAATCACCACCCACAG		
Cytochrome–RTF	FJ594967	TGAGCCGCAACAGTAATC	Real-time PCR	80
Cytochrome–RTR		AAGGGAAAAGGAAGTGAAAG		

**Table 2 foods-14-01404-t002:** Sequences for multiple sequence alignment.

Abbreviation	Species	Accession No.
CYT_CHIJI	*Chilobrachysguangxiensis*	B1P1J3
CYTL_TACTR	*Tachypleustridentatus*	Q7M429
CYTC_MACMU	*Macaca mulatta*	O19092
CYTC_SAISC	*Saimiri sciureus*	O19093
CYTC_RAT	*Rattus norvegicus*	P14841
CYTC_MOUSE	*Mus musculus*	P21460
CYTN_HUMAN	*Homo sapiens*	P01037
CYT_CHICK	*Gallus gallus*	P01038
CYT1_CROAD	*Crotalus adamanteus*	J3RYX9
CYT_COTJA	*Coturnix japonica*	P81061
CYTC_RABIT	*Oryctolagus cuniculus*	O97862
CYTT_HUMAN	*Homo sapiens*	P09228

## Data Availability

The original contributions presented in the study are included in the article, further inquiries can be directed to the corresponding author.

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
