# Peer review of "A Novel Cystatin Gene from Sea Cucumber (Apostichopus japonicus): Characterization and Comparative Expression with Cathepsin L During Early Stage of Hypoxic Exposure-Induced Autolysis"

_foods, 2025, doi:10.3390/foods14081404_

Round 1

Reviewer 1 Report

Comments and Suggestions for Authors

The authors provide an elaborated account of the cystatin gene (SjCyt) cloned from (Api)Stichopus japonicus. The study includes its genetic structure, expression at the transcriptomic and translational level, localization in the tissues by immunofluorescence and even recombinant cloning to obtain purified protein. The study provides interesting and valuable data about the dynamics of this cystatin and cathepsins in sea urchin. However, there are quite a few points to be addressed, before potential publication. Please see my specific comments:

Line 2: the species name. It seems that: This species is now formally accepted within the monotypic genus Apostichopus and named A. japonicus in the World Register of Marine Species (Paulay, 2013). Please adjust accordingly and/or explain it in the text

Line 10 a threat

Line 21 at early

Line 25 SjCyt and

Line 36 – 40 please provide some reference for this

Lines 41 – 44 The data provided here are referring to aquaculture output, however the link provided lists only wild capture data, where there is no data for China. I think the correct link might be: https://www.fao.org/fishery/statistics-query/en/aquaculture/aquaculture_quantity

The numbers you provided can also be obtained through FAO's FishStatJ software (https://www.fao.org/fishery/en/statistics/software/fishstatj), the FAO Global Fishery and Aquaculture Production Statistics worksheet. If any of these are the correct source, please include it and cite according to the instructions by FAO.

Lines 51 – 53 This sentence needs to be grammatically corrected, there is something missing

Line 56 – 57 matrix metalloproteases – please adjust correctly throughout the text

Line 58 sentence not grammatically correct

Line 59 is strictly controlled

Line 61 in autolysis

Lines 63 – 68 please rephrase this part. Although the source reference has been included, the text is almost identical

Line 68 please use plural when referring to the entire family of enzymes, like cystatins. Apply this throughout the text for other families too

Lines 94 – 97 At the end of introduction you mention that cucumbers were captured without ultraviolet exposure. What does this mean? Does this mean the experiment was conducted in the dark? By night? Please specify so in the materials and methods section. Please also make sure you properly explain the entire design and specify exactly how many replicates were included for each experimental group.

Lines 99 – 103 please specify which primers were used for the reverse transcription and how much input RNA. Was the integrity of extracted RNA verified in any way?

Lines 104 – 108 please specify exactly which GenBank sequences were used for the primer design, and update the Table 1 with primer characteristics, like annealing temperature and expected product size. For the Real-time PCR primers also provide their efficiency. Otherwise, it is very difficult if not impossible to reuse these primers

Lines 111 – 112 which database was used for tblastx at NCBI?

Line 116 here it is mentioned that Neighbor-Joining tree was constructed, however in the results it states Maximum likelihood (line 258). I suggest to use maximum likelihood as a more evolutionary informed method,  and correct accordingly

Line 118 please specify what kind of computer simulation model was generated by I-TASSER, this is very vague

Line 120 which sequences of SjCyt−cathepsin L and SjCyt−cathepsin B were used for this?

Line 146 synthesised

Lines 146 – 155 Were RT-qPCR reactions conducted in duplicates/triplicates? Please specify exactly how many replicates are for each experiment

Line 155 The  2−ΔΔCt method can only be used if the efficiency of the primers is > 90% (> 1.9). Please provide this information for every primer, as in previous comments, and adjust the calculation if necessary (use real efficiency instead of 2)

Lines 162 – 163 According to results, you also used beta actin as a reference in western blot. Please provide information about its antibody. Again, please specify the number of replicates, if any.

Line 173 immersion of the sections in

Lines 194 – 195 please provide more details about the transformation process. It is transformation rather than transfection when using bacterial cells

Lines 218 – 222 Statistical analyses is the most problematic part of the study. Using ANOVA requires that the data follow normal distribution and display equal variances. With only three replicates per group this cannot be adequately ascertained. In my opinion you can:

  • Assess the distribution of the residuals in an ANOVA model (probably will need another software)
  • Use non-parametric test, like Mann Whitney. This would be the best option. It seems many between group comparisons were made (since post-hoc tests are not mentioned that usually follow an ANOVA analyses, and the ANOVA results per se are not reported), so there is no added value from using a test that compares all groups at once

AND very importantly, when there are many between group comparisons, regardless of the test, a multiple testing correction for the p value needs to be applied. The probability of a false positive is 5% for each comparison, which means it increases with every comparison made

Line 223 although there are some elements of the discussion in the results, as you have a separate discussion section, I suggest naming paragraph 3 as only Results

Line 230 What is SMART analyses, it is not mentioned in the Materials and methods section. Need to specify full name upon the first mention.

Line 259 please provide data for all the 22 sequences used in the phylogenetic tree, especially their accession numbers and species. This can be a supplementary file.  

Line 270 Many results are shown in figure 2, but its resolution is so low I cannot judge on its content, parts A and C are particularly unreadable. This needs to be corrected.

Please also specify what different colors describe in the phylogenetic tree.

Line 343 form?

Line 351 In contrast?

Lines 361 How did you check the stability of cytb as a reference gene?

Line 362 and Figure 7   So the data were standardized by the expression of cytb gene and then all divided with the expression in longitudinal muscle. This was done for tissue expression (Figure 7 A and B), but what did you normalize it against, except cytb, for the temporal expression (Figure 7 C and D). That is not clear nor mentioned.

This is another problem for the ANOVA statistical analyses. This means that what you are comparing are differences in ratios (fold changes which is expected in qPCR), but ratios inherently do not follow normal distribution – you do not even need to check for this.

I suggest applying a log transformation to this data (after standardization to reference gene) which stabilizes the variances and makes data linear. Plus in the log space when you compare the differences in means (which ANOVA does) you actually compare ratios in normal space, completely in accordance with the nature of this data. Log2 works very well for expression data. So there is no need to divide by the control group for the statistical test.

Without some kind of  transformation, these data are not acceptable for classical ANOVA test.

Line 365 least

Line 396 Figure 8 same comment as above for qPCR data

Is it not the expression in tube feet shown in A and B (according to markings on western blot gel), and dorsal epidermis in C and D? The inverse is stated in the figure legend.

Line 435 Figure 9 this figure is in such a poor resolution I cannot judge any results related to it. This needs to be corrected, with the scale bar and magnification clearly stated. Also please use arrows to designated specific features on the figures, like the peri−nucleus SjCyt−SjCL complex, to name only one. If this is not possible on a smaller magnification, please provide at least one image with higher magnification and resolution.

Also, please better explain what is shown in Figure 9C. I sort of have the idea, but not sure. Also, this seems to be count data, all normal distribution issues apply here as well. Please specify how many replicates are behind each bar.

Line 446 was purified rSjCyt confirmed by western blot?

The marker on images (fig 10) goes down to 14.4 kDa, and it is stated the molecular weight of purified rSjCyt was 13 kDa – so either this band is not shown, or the molecular weight is not correct? Which band is the target?

Also, in figure 10C, do you have any idea what comprises the 200 kDa band, whose degradation was attenuated by the addition of cystatin? Conversely, the degradation of lower bands (at approx. 116 and 97.2) was increased. Any explanation for this? Please comment in the discussion

Line 461 the Discussion requires overall English language proofreading and grammatical corrections. There are many sentences that are hard to understand due to this.

Also, please refer to the phylogeny of sea urchin cystatin in the discussion, this part was not covered.

Comments on the Quality of English Language

Please see main comments. English language needs revising, especially the discussion.

Author Response

1)Line 2: the species name. It seems that: This species is now formally accepted within the monotypic genus Apostichopus and named A. japonicus in the World Register of Marine Species (Paulay, 2013). Please adjust accordingly and/or explain it in the text

Reply: We are very grateful for pointing this out. We have replaced stichopus with Apostichopus.

2)Line 10 a threat

Reply: This has been corrected (See line10 for details).

3)Line 21 at early

Reply: This has been corrected (See line 21 for details).

4)Line 25 SjCyt and

Reply: This has been corrected (See line 25 for details) .

5)Line 36 – 40 please provide some reference for this

Reply: Thank you for your suggestion. It not very correct to stating “The commercial price of Stichopus japonicus has long been standing the first place on the list of marine foods in China”. So we make some modification “In recent 20 years, ApoStichopus japonicus has consistently maintained high market valuea mong all marine products in China”. We believe that this merely represents the current situation regarding the popularity of the sea cucumber and does not necessitate support from references.

6)Lines 41 – 44 The data provided here are referring to aquaculture output, however the link provided lists only wild capture data, where there is no data for China. I think the correct link might be: https://www.fao.org/fishery/statistics-query/en/aquaculture/aquaculture_quantity

The numbers you provided can also be obtained through FAO's FishStatJ software (https://www.fao.org/fishery/en/statistics/software/fishstatj), the FAO Global Fishery and Aquaculture Production Statistics worksheet. If any of these are the correct source, please include it and cite according to the instructions by FAO.

Reply: Thank you for pointing this out, and we have used the above recommended link https://www.fao.org/fishery/statistics-query/en/aquaculture/aquaculture_quantity. (See line 44 for details)

7)Lines 51 – 53 This sentence needs to be grammatically corrected, there is something missing

Reply: The errors have been corrected.

8)Line 56 – 57 matrix metalloproteases – please adjust correctly throughout the text

Reply: The “matrix metal protease” has been replaced with matrix metalloproteases.

9) Line 58 sentence not grammatically correct

Reply: The error syntax has been corrected .

10)Line 59 is strictly controlled

Reply: This has been corrected.

11)Line 61 in autolysis

Reply: This has been corrected.

12) Lines 63 – 68 please rephrase this part. Although the source reference has been included, the text is almost identical.

Reply: This part has been rephrased as below.

Before correction: It is reported that biosynthesis of cysteine protease is regulated by compartmentalization to lysosomes, activation of proenzyme and their endogenous protein inhibitors [5]. Among them the best characterized are cystatins, which function as reversible, tight−binding inhibitors of cysteine proteases. Cystatins comprise a super−family of evolutionary related proteins, each consisting of at least one domain of 100−120 amino acid residues with conserved sequence motifs [5].

After correction: Research indicates that the biosynthesis of cysteine proteases is controlled through multiple mechanisms, including lysosomal compartmentalization, proenzyme activation, and endogenous protein inhibitors [5]. One of the most well-studied inhibitor families is the cystatins, which act as highly specific, reversible inhibitors of cysteine proteases. Cystatins comprise a large superfamily of evolutionarily conserved proteins, each containing at least one structural domain of approximately 100–120 amino acids with characteristic sequence motifs [5].

13)Line 68 please use plural when referring to the entire family of enzymes, like cystatins. Apply this throughout the text for other families too

Reply: This has been corrected (See line 66 for detail).

14)Lines 94 – 97 At the end of introduction you mention that cucumbers were captured without ultraviolet exposure. What does this mean? Does this mean the experiment was conducted in the dark? By night? Please specify so in the materials and methods section. Please also make sure you properly explain the entire design and specify exactly how many replicates were included for each experimental group.

Reply: This is not a correct expression. Some previous studies used ultraviolet lamp to strengthen the stimulation to induce sea cucumber autolysis. In the last paragraph of introduction, I want to emphasize that in this study we didn’t use ultraviolet lamp on purpose. Actually it is not necessary to mention ultraviolet exposure, it has nothing to do with this study. So we have made some modifications at the end of introduction.

We have conducted studies on sea cucumber autolysis for over 13 years. We know the individual difference is often significant. So we increased the sample numbers, 15 sea cucumber as replicate were used for gene distribution in different tissues, 9 sea cucumbers as replicate were used at each time point for gene temporal expression. Additions were made in Material 2.1.

15)Lines 99 – 103 please specify which primers were used for the reverse transcription and how much input RNA. Was the integrity of extracted RNA verified in any way?

Reply: Thank you for pointing this out. During reverse transcription, each system contained 2 μg of RNA. In addition, the integrity of the RNA was verified by agarose gel electrophoresis, in which the 18S and 28S ribosomal RNA bands had good integrity.

16)Lines 104 – 108 please specify exactly which GenBank sequences were used for the primer design, and update the Table 1 with primer characteristics, like annealing temperature and expected product size. For the Real-time PCR primers also provide their efficiency. Otherwise, it is very difficult if not impossible to reuse these primers

Reply: The Genbank sequences used for our primer design were just the genes that listed in table 2, which were used for amino acid homology comparisons.

We have added the Tm values and GenBank number of the primers in RT-qPCR in Table 1. And the amplification efficiencies of the other three sets of primers were all greater than 90%. The expected PCR product is approximate in a range of 200-300bp, we didn’t add it into table 1. The efficiency of the real-time PCR primers was provided as below.

(1) Quantification standard curve for reference gene Cytb amplification

The RT-PCR results for Cytb are presented in Fig. 2-4. The figures demonstrated the stable Ct values and fluorescence intensity throughout the PCR amplification. Melting curve showed a single specific peak with no primer-dimers or non-specific amplification products, exhibiting a Tm of 80.25 °C. The standard curve was established by plotting the template quantity against Ct values, yielding the equation Y = -3.433log(x) + 27.684. The amplification efficiency was calculated to be 0.956 (95.6%).

Fig.2 Amplification curve of reference gene Cytb

Fig.3 Melting curve of reference gene Cytb

Fig.4 Standard curve of Cytb template quantity versus CT values

(2) Quantification standard curve for SjCyt gene amplification

The RT-PCR results for SjCyt are presented in Fig. 5-7. The figures demonstrated the stable Ct values and fluorescence intensity throughout the PCR amplification. Melting curve showed a single specific peak with no primer-dimers or non-specific amplification products, exhibiting a Tm of 79.91 °C. The standard curve was established by plotting the template quantity against Ct values, yielding the equation Y=-3.253log(x)+35.155. The amplification efficiency was calculated to be 1.030 (103.0%).

Fig.5 Amplification curve of SjCyt gene

Fig.6 Melting curve of SjCyt

Fig.7 Standard curve of SjCyt template quantity versus CT values

(3) Quantification standard curve for SjCL gene amplification

The RT-PCR results for SjCL are presented in Fig. 8-10. The figures demonstrated the stable Ct values and fluorescence intensity throughout the PCR amplification. Melting curve showed a single specific peak with no primer-dimers or non-specific amplification products, exhibiting a Tm of 84.46 °C. The standard curve was established by plotting the template quantity against Ct values, yielding the equation Y=-3.276log(x)+31.175. The amplification efficiency was calculated to be 1.019 (101.9%).

Fig.8 Amplification curve of SjCL

Fig.9 Melting curve of SjCL

Fig.10 Standard curve of SjCL template quantity versus CT values

17) Lines 111 – 112 which database was used for tblastx at NCBI?(Non - redundant protein sequences)

Reply: Yes, non-redundant protein sequences were used for tblastx and this was added to the manuscript.

18)Line 116 here it is mentioned that Neighbor-Joining tree was constructed, however in the results it states Maximum likelihood (line 258). I suggest to use maximum likelihood as a more evolutionary informed method, and correct accordingly

Reply: Thank you for pointing this out. This expression line 258 is a linguistic misrepresentation. Yes, maximum likelihood is a more evolutionary informed method, but Neighbor-Joining tree is efficient enough to construct the phylogenetic tree and illustrate. We have replaced Maximum likelihood in line 258 with Neighbor-Joining. If you do insist on Maximum likelihood, we will make it later.

19)Line 118 please specify what kind of computer simulation model was generated by I-TASSER, this is very vague

Reply: Thank you for catching this oversight. Additional information about the model was added (see section 2.3 for details).

20)Line 120 which sequences of SjCyt−cathepsin L and SjCyt−cathepsin B were used for this?

Reply: We have added Uniprot accession number after cathepsin L and cathepsin B.

21) Line 146 synthesised

Reply: This mistake has been corrected as synthesized. (See line 155 for detail)

22)Lines 146 – 155 Were RT-qPCR reactions conducted in duplicates/triplicates? Please specify exactly how many replicates are for each experiment

Reply: Thank you for pointing this out. We repeated this three times for each cDNA template. In the RT-qPCR experiments for tissue distribution analysis, tissues were collected from randomly selected 15 living sea cucumbers. These sea cucumbers were grouped into three groups of five individuals each. Regarding the temporal RT-qPCR experiment, 63 living sea cucumbers was randomly captured. They were then divided into three groups. Three sample was collected from each group at distinct time intervals.  

22’) Line 155 The 2−ΔΔCt method can only be used if the efficiency of the primers is > 90% (> 1.9). Please provide this information for every primer, as in previous comments, and adjust the calculation if necessary (use real efficiency instead of 2)

Reply: Thank you for pointing this out. As shown in previous comment 16, the efficiency of all the primers are more than 90% (Cytb is 95.6%; SjCyt is 103.0%; SjCL is 101.9%), so the method of 2-ΔΔCt could be used for qualification. We validated the ∆∆Ct method for the reference gene Cytb and SjCyt, and obtained linear regression equations with slopes close to 0, proving that the available ∆∆Ct analysis method. Details are presented below.

(a) Validation the ∆∆Ct method for Cytb and SjCyt

The obtained linear regression equation, y =- 0.0827Log(X) +7.3871, has a slope of -0.0827, which is close to 0. This proves that the ∆∆Ct method can be used to analyze the expression of SjCyt using Cytb as reference gene.

Fig. 11 Verification △△Ct method between Cytb and SjCyt

(b) Validation the ∆∆Ct method for Cytb and SjCL

The linear regression equation y =- 0.0187Log(X) +4.0503, with a slope of -0.0187, which is close to 0. This proves that the expression of the SjCL gene can be quantified by the △△Ct method, using Cytb as the internal reference gene.

Fig.12 Verification △△Ct method between Cytb and SjCL

23)Lines 162 – 163 According to results, you also used beta actin as a reference in western blot. Please provide information about its antibody. Again, please specify the number of replicates, if any.

Reply: Thank you for pointing this out. We have added the antibody information in the article. (See line 172 for detail). In addition, our experiment was repeated three times.

24) Line 173 immersion of the sections in

Reply: Thank you, your correction was used.

25)Lines 194 – 195 please provide more details about the transformation process. It is transformation rather than transfection when using bacterial cells

Reply: Thank you for your kind correction. We are following the instructions for the chemical transformation method, and we add description of the conversion method, and “transfection” was revised to transformation.

26) Lines 218 – 222 Statistical analyses is the most problematic part of the study. Using ANOVA requires that the data follow normal distribution and display equal variances. With only three replicates per group this cannot be adequately ascertained. In my opinion you can:

Assess the distribution of the residuals in an ANOVA model (probably will need another software)

Use non-parametric test, like Mann Whitney. This would be the best option. It seems many between group comparisons were made (since post-hoc tests are not mentioned that usually follow an ANOVA analyses, and the ANOVA results per se are not reported), so there is no added value from using a test that compares all groups at once

AND very importantly, when there are many between group comparisons, regardless of the test, a multiple testing correction for the p value needs to be applied. The probability of a false positive is 5% for each comparison, which means it increases with every comparison made.

Reply: Thank you for your patient and kind suggestion. According to your comments, we log2 transformed the data in C and D of Figure 7. And, we performed normal distribution and variance chi-square tests using SPASS before performing one-way ANOVA, which proved that the data were consistent with normal distribution and variance chi-square.

27)Line 223 although there are some elements of the discussion in the results, as you have a separate discussion section, I suggest naming paragraph 3 as only Results

Reply: Thank you for pointing this out. We have changed “3. Results and Discussion” to “3. Results”.

28)Line 230 What is SMART analyses, it is not mentioned in the Materials and methods section. Need to specify full name upon the first mention.

Reply: Full name of SMART and its websit have been added to Material 2.3.

29)Line 259 please provide data for all the 22 sequences used in the phylogenetic tree, especially their accession numbers and species. This can be a supplementary file.

Reply: The names, accession numbers and specific sequences of the 22 species were provided in the supplementary file.

30)Line 270 Many results are shown in figure 2, but its resolution is so low I cannot judge on its content, parts A and C are particularly unreadable. This needs to be corrected.

Reply: The image resolution was increased and zoomed in on figure 2A and 2C.

31)Please also specify what different colors describe in the phylogenetic tree.

Reply: A note of colors describe was added in the capture of figure 2.

32)Line 343 form?

Reply: Yes, it was an spelling error, which has been corrected in the body of the text.

33)Line 351(362) In contrast?

Reply: We want to express “the following is different, even on the contrary, comparing with previous description”.

34)Lines 361 How did you check the stability of cytb as a reference gene?

Reply: Rreference gene, cytb, has been used for years. Its stability was checked by many replication assays.

35)Line 362 and Figure 7 So the data were standardized by the expression of cytb gene and then all divided with the expression in longitudinal muscle. This was done for tissue expression (Figure 7 A and B), but what did you normalize it against, except cytb, for the temporal expression (Figure 7 C and D). That is not clear nor mentioned.

This is another problem for the ANOVA statistical analyses. This means that what you are comparing are differences in ratios (fold changes which is expected in qPCR), but ratios inherently do not follow normal distribution you do not even need to check for this.

I suggest applying a log transformation to this data (after standardization to reference gene) which stabilizes the variances and makes data linear. Plus in the log space when you compare the differences in means (which ANOVA does) you actually compare ratios in normal space, completely in accordance with the nature of this data. Log2 works very well for expression data. So there is no need to divide by the control group for the statistical test.

Without some kind of transformation, these data are not acceptable for classical ANOVA test.

Reply: Thank you so much for this suggestion. For temporal expression, the data was standardized against the expression level at 0 h . We took your suggestion and performed a Log2 transformation of the relative expressions and again performed a one-way ANOVA.

36) Line 365 least

Reply: We have corrected the grammar mistakes.

37) Line 396 Figure 8 same comment as above for qPCR data. Is it not the expression in tube feet shown in A and B (according to markings on western blot gel), and dorsal epidermis in C and D? The inverse is stated in the figure legend.

Reply: Thank you so much for pointing this. Changes have been made to the capture in Figure 8. (See line 441-413 for details) In addition, we have reversed the top and bottom order of AB and CD in Figure 8.

38)Line 435 Figure 9 this figure is in such a poor resolution I cannot judge any results related to it. This needs to be corrected, with the scale bar and magnification clearly stated. Also please use arrows to designated specific features on the figures, like the peri−nucleus SjCyt−SjCL complex, to name only one. If this is not possible on a smaller magnification, please provide at least one image with higher magnification and resolution.

Also, please better explain what is shown in Figure 9C. I sort of have the idea, but not sure. Also, this seems to be count data, all normal distribution issues apply here as well. Please specify how many replicates are behind each bar.

Reply: We have enhanced the image resolution of Fig. 9 with an arrow pointing to the peri-nucleus SjCyt-SjCL complex. We performed three replicate counts behind each graph, and Figure 9C represents the counting statistics for the fluorescent sites.

39)Line 446 was purified rSjCyt confirmed by western blot?

Reply: The purified rSjCyt protein are confirmed by SDS-PAGE, and western blot validation was conducted (image was not supplied in the article).

40)The marker on images (fig 10) goes down to 14.4 kDa, and it is stated the molecular weight of purified rSjCyt was 13 kDa – so either this band is not shown, or the molecular weight is not correct? Which band is the target?

Reply: Molecular weight of SjCyt is predicted to to be 13kDa. The recombinant SjCyt (rSjCyt) carried tag of 6×his, so the total molecular weight of rSjCyt is: 13kDa+0.9kDa≈14kDa.

41)Also, in figure 10C, do you have any idea what comprises the 200 kDa band, whose degradation was attenuated by the addition of cystatin? Conversely, the degradation of lower bands (at approx. 116 and 97.2) was increased. Any explanation for this? Please comment in the discussion.

Reply: We believe that the 116 and 97.2 KDa protein bands are degradation product of 200 kDa protein, and the addition of SjCyt can inhibit such degradation. We did not identify the protein of 200 kDa, this might be our future work .

42)Line 461 the Discussion requires overall English language proofreading and grammatical corrections. There are many sentences that are hard to understand due to this.

Reply: Thank you so much for your suggestion. The English language has been improved.

43)Also, please refer to the phylogeny of sea urchin cystatin in the discussion, this part was not covered.

Reply: Thank you so much for your valuable suggestion. Sea urchins and sea cucumbers share the closest phylogenetic relationship in echinoderm phylogeny. Unfortunately, no studies on cystatin of sea urchin could be found.

We sincerely appreciate the reviewers' meticulous and constructive comments.

Reviewer 2 Report

Comments and Suggestions for Authors

Dear authors,

I appreciate your work and the presentation of the results achieved. However, there are a few minor revisions that need to be made before publishing:

  • missing gaps in Abstract
    line 14: (SjCyt)belonging
    line 25:  SjCytand
    line 26:  recombinantSjCyt
  • line 61: please correct/rewrite the part of the sentence "protease inhibitor o autolysis. " (inhibitor in autolysis)
  • lines 107, 108 (as well as throughout the following text): Please unify the style and add the country to "Takara" similar to the previously mentioned companies/producers. 
  • Table 1: please correct " List of primer sequence used in this study " to "List of primer sequences used in this study "
  • line 146: "syntheised" to "synthesised"
  • lines 226-228:  I think that in the sentence "Analysis revealed that the complete cDNA of SjCyt contained a 5’ untranslated region of 22 bp and an open reading frame of 348 bp, and the open reading frame was 348 bp in length, encoding 115 amino acids." one mention of the open reading frame should be deleted.
  • Check the text and add a space if necessary. For example: "DynamicsSimulations" (line 119), "containeda" (line 227), "stateof" (line 307) etc.
  • Please provide Figures 2a and 2c in better resolution to make the text more readable.
  • Figure 8: is it possible to use a larger font for graphs A-D?
  • Figure 9: I recommend trying to increase the contrast in images in series A and B; the fluorescence signal is not very visible.

Author Response

1)line 14: (SjCyt) belonging

Reply: Thank you, this mistake has been corrected and highlighted in the article.

2) line 25: SjCytand

Reply: Thank you, it has been corrected.

3)line 26: recombinantSjCyt

Reply: Thank you, it has been corrected.

4)line 61: please correct/rewrite the part of the sentence "protease inhibitor o autolysis. " (inhibitor in autolysis)

Reply: Thank you! “inhibitor o autolysis” has been change to “inhibitor in autolysis”.

5)lines 107, 108 (as well as throughout the following text): Please unify the style and add the country to "Takara" similar to the previously mentioned companies/producers.

Reply: Thank you for pointing this out. We have added the country of Japan to "Takara".

6)Table 1: please correct " List of primer sequence used in this study " to "List of primer sequences used in this study "

Reply: Thank you, it has been corrected.

7) line 146: "synthesised" to "synthesised"

Reply: Thank you, “synthesized” is the correct one.

8) lines 226-228: I think that in the sentence "Analysis revealed that the complete cDNA of SjCyt contained a 5’ untranslated region of 22 bp and an open reading frame of 348 bp, and the open reading frame was 348 bp in length, encoding 115 amino acids." one mention of the open reading frame should be deleted.

Reply: Thank you for pointing this out. This was indeed an error, and corrected.

9)Check the text and add a space if necessary. For example: "DynamicsSimulations" (line 119), "containeda" (line 227), "stateof" (line 307) etc.

Reply: Thank you! These error has been corrected.

10)Please provide Figures 2a and 2c in better resolution to make the text more readable.

Reply: We have improved the clarity of Figures 2 and zoomed in on A and C.

11)Figure 8: is it possible to use a larger font for graphs A-D?

Reply: Thank you! The font of Figure 8 has been raised.

12)Figure 9: I recommend trying to increase the contrast in images in series A and B; the fluorescence signal is not very visible.

Reply: The contrast of Figure 9 has been increased.

Round 2

Reviewer 1 Report

Comments and Suggestions for Authors

Dear authors,

Thank you for making significant changes to the manuscript and clarifying many points that were confusing in the first version. I accept your reasoning and explanations. Many comments have been made in the cover letter; however not all were included in the submitted revised version of the manuscript. And they should be included for the benefit of future readers.

My specific comments are:

Line 41 'During that time' instead of 'In recent 20 years' – just to avoid using the latter expression twice in the same paragraph

Line 46 please include the new link as you mention in your comments, the old one is still in the manuscript. The new one is:

https://www.fao.org/fishery/statistics-query/en/aquaculture/aquaculture_quantity

Line 56 maybe use: 'like collagen' – to improve readability of the sentence

Line 60 – 61 the confusing, grammatically incorrect, sentence remained the same, please rephrase the sentence: 'Until now, almost all the studies of autolysis focused the mechanism on the endogenous protease throughout the sea cucumber.'

Lines 66 – 68 changes indicated in author response have not been included in the manuscript, please include them:

After correction: Research indicates that the biosynthesis of cysteine proteases is controlled through multiple mechanisms, including lysosomal compartmentalization, proenzyme activation, and endogenous protein inhibitors [5]. One of the most well-studied inhibitor families is the cystatins, which act as highly specific, reversible inhibitors of cysteine proteases. Cystatins comprise a large superfamily of evolutionarily conserved proteins, each containing at least one structural domain of approximately 100–120 amino acids with characteristic sequence motifs [5].

Line 84 Thank you for explaining what is meant by UV exposure. I agree, there is no need to specifically mention that here at all, or just briefly mention it. Please rephrase this sentence in the style:

Therefore, in this study, the adult living sea cucumbers were captured and autolysis experiments were conducted under hypoxic conditions, without additional UV stimulation as done in other studies (HERE PUT REFERENCES ON SEVERAL OF THOSE STUDIES).

Line 88 delete systematic

Line 94 - 104 – thank you for clarifying the experimental design. There are enough replicates for each analysis, that is very good, however the separation of individuals into three groups is a bit confusing. I understand it is sort of a 'replication of the experiment', but with tissue distribution assay, there is no real experiment, it just 15 individuals analyzed, correct? Unless they were somehow processed in batches of five and you think this might have influenced the result?

Same goes for time course hypoxia experiment – this is indeed an experiment, and the repetition of the experiment is always welcomed, but if they were all processed at the same time, then it is not really repetition. Additionally, if samples were split into these groups, then in statistical analyses you need to use nested ANOVA, because then you have between group variation within each experimental condition. So these two things need to be in line. In my opinion omitting the mention of the three groups in experimental design does not hurt the study outcome, and the data were analyzed in this way anyway. However, if you do decide to mention the three groups, then please adjust the statistical analyses accordingly.

Line 110 Thank you for providing the information about the input amount of RNA, please also provide which primers were used for first strand cDNA synthesis, oligo-dT, random hexamers, both, or gene specific? The same goes for qPCR.

Aso, add the sentence: the integrity of the RNA was verified by agarose gel electrophoresis, in which the 18S and 28S ribosomal RNA bands had good integrity.

Line 120 TBLASTX requires a nucleotide query sequence and a nucleotide database, not a protein database. Perhaps you mean non-reduntant nucleotide database at NCBI (nr/nt)? Or was it a different kind of BLAST?

Lines 127 – 128 a verb is missing in this sentence

Lines 161 – 171 Thank you for clarifying in your comments the determination of primer efficiency. I have no doubt you have done a good job, however this needs to be reported, even briefly, for the purpose of scientific transparency and replicability, in case other researchers wish to use your primers and assays. In this section a line could be inserted/added, such as:

‘The amplification efficiencies of of primers were greater than 90% (Cytb 95.6%; SjCyt 103.0%; SjCL 101.9%). The expected PCR product was in the range of 200-300 bp. All reactions were conducted in triplicates.’

Line 211 transformed

Line 212 What does it mean by chemical method? Which chemical? Please specify.

Line 236 - 240  Statistical analyses

When you mention the three groups here the way they are now, it is confusing. No need to explain experimental design here, as it was already done in materials and methods. Also, all that you have done with the data needs to be reported here. Thank you for performing the log2 transformation on qPCR time course data, but it needs to be performed for all qPCR data. Even western blot – as it was standardized to beta actin? Actually, not sure, it depends on normality tests.

Reading in retrospect my comment, I see how it might have been misunderstood to lead you to do it only for part of the data. All that you have analyzed with ANOVA, first you need to check normality and equal distribution of variances – ALL experimental data and then apply a transformation if necessary, and it needs to be mentioned in this section.

Also, the authors did not address the problem of multiple testing corrections. You have made many comparisons and the p value obtained must be corrected for this. Bonferroni or Benjamini-Hochberg corrections are usually used. Please do so and mention this in the statistical analyses section.

So this section could be something like:

The statistical analyses of the data was performed by using one-way ANOVA in GraphPad Prism 9.5 (Graphpad software Inc., San Diego,CA, USA). The normal distribution and variance chi-square tests using SPASS were performed to ensure the data met ANOVA assumptions. The qPCR data were log2 transformed. The data were represented as mean±standard deviation, and the p < 0.05 was considered as statistically significant after correction ….?

Line 293 Thank you providing a much better image. It is clear now.

The designation of colors for specific cystatin families in the figure is wrong, is it not blue for kininogen? Please check. Also, cystatins

Line 379 So here perhaps you could reference one of those assays were it was proven cytb was a suitable housekeeping gene?

Line 385 The legend is missing in Figure C and D.

Please do not forget to log2 transform data for A and B as well.

Line 394 Here it states the temporal data was normalized against 0 hours, but it is not very clear. Please expand the explanation. You could also add this in the figure caption.

Line 454 Figure 9. Thank you for improving this image. Please add in figure caption what does the arrow mean. Also designate some other features that might be important.

It is still not exactly clear what Figure 9C shows, please explain how was these quantitative analyses of the immunofluorescence performed. What are the pluses and the minuses, etc. It could fit in the figure caption.

Line 462 please do mention here that: The purified rSjCyt protein was confirmed by SDS-PAGE, and western blot validation was conducted (data not shown).

Thank you for improving the discussion. Please just check the grammar after track changes are removed, I think there are few that remained.

Comments on the Quality of English Language

In the main comments.

Author Response

Dear Reviewer,

I sincerely appreciate your meticulous review and friendly suggestions. Through our communication, I've not only recognized my own deficiencies but also acquired a wealth of professional knowledge. I am honored to have you as my reviewer!

In accordance with your valuable comments, we have made corresponding revisions to the manuscript. Moreover, we have conducted a thorough check of the details, which are presented as follows:

(1) Line 41 'During that time' instead of 'In recent 20 years' – just to avoid using the latter expression twice in the same paragraph.

Reply: Thanks for your suggestion, it has been changed in the manuscript. (See line 41 for details)

(2) Line 46 please include the new link as you mention in your comments, the old one is still in the manuscript. The new one is: https://www.fao.org/fishery/statistics-query/en/aquaculture/aquaculture_quantity

Reply: Thanks for your suggestion, it has been changed in the manuscript. (See detail for line 45)

(3) Line 56 maybe use: 'like collagen' – to improve readability of the sentence

Reply: Thanks, “collagen” was replaced by “like collagen”. (See detail for line 55).

(4) Line 60 – 61 the confusing, grammatically incorrect, sentence remained the same, please rephrase the sentence: 'Until now, almost all the studies of autolysis focused the mechanism on the endogenous protease throughout the sea cucumber.'

Reply: We apologize for the grammatical error we had and have corrected the sentence errors:

At the present time, almost all the studies on the autolysis of sea cucumber have mainly focused on endogenous proteases. (See details for line 59-61).

(5) Lines 66 – 68 changes indicated in author response have not been included in the manuscript, please include them:

After correction: Research indicates that the biosynthesis of cysteine proteases is controlled through multiple mechanisms, including lysosomal compartmentalization, proenzyme activation, and endogenous protein inhibitors [5]. One of the most well-studied inhibitor families is the cystatins, which act as highly specific, reversible inhibitors of cysteine proteases. Cystatins comprise a large superfamily of evolutionarily conserved proteins, each containing at least one structural domain of approximately 100–120 amino acids with characteristic sequence motifs [5].

Reply: Thank you for your suggestions, we have made additions. (See details for 65-71)

(6) Line 84 Thank you for explaining what is meant by UV exposure. I agree, there is no need to specifically mention that here at all, or just briefly mention it. Please rephrase this sentence in the style:

Therefore, in this study, the adult living sea cucumbers were captured and autolysis experiments were conducted under hypoxic conditions, without additional UV stimulation as done in other studies (HERE PUT REFERENCES ON SEVERAL OF THOSE STUDIES).

Reply: Thank you for your suggestion, we have changed it in the manuscript:

Therefore, in this study, the adult living sea cucumbers were captured and autolysis experiments were conducted under hypoxic conditions, without additional UV stimu-lation as done in other studies [1,2] (See details for line 82-84)

(7) Line 88 delete systematic

Reply: Thank you for your suggestions, we have deleted it. (See detail for line 87)

(8) Line 94 - 104 – thank you for clarifying the experimental design. There are enough replicates for each analysis, that is very good, however the separation of individuals into three groups is a bit confusing. I understand it is sort of a 'replication of the experiment', but with tissue distribution assay, there is no real experiment, it just 15 individuals analyzed, correct? Unless they were somehow processed in batches of five and you think this might have influenced the result?

Same goes for time course hypoxia experiment – this is indeed an experiment, and the repetition of the experiment is always welcomed, but if they were all processed at the same time, then it is not really repetition. Additionally, if samples were split into these groups, then in statistical analyses you need to use nested ANOVA, because then you have between group variation within each experimental condition. So these two things need to be in line. In my opinion omitting the mention of the three groups in experimental design does not hurt the study outcome, and the data were analyzed in this way anyway. However, if you do decide to mention the three groups, then please adjust the statistical analyses accordingly.

Reply: I’m very sorry that the explanation for this question is not clear. As you mentioned, in the tissue distribution experiment, we selected a total of 15 sea cucumbers. Specifically, we selected 5 sea cucumbers in each batch, and the experiment was conducted in three times. We made changes in the manuscript:

Line 93: Randomly selected 5 living sea cucumbers

Line 102 (last sentence of this experimental method): Conduct each experiment three times, and each repetition should be an independent trial.

(9) Line 110 Thank you for providing the information about the input amount of RNA, please also provide which primers were used for first strand cDNA synthesis, oligo-dT, random hexamers, both, or gene specific? The same goes for qPCR

Aso, add the sentence: the integrity of the RNA was verified by agarose gel electrophoresis, in which the 18S and 28S ribosomal RNA bands had good integrity.

Reply: We looked up the instructions for the reverse transcription kit to synthesize the cDNA, and the primers used Random 6 mers and Oligo dT Primer. Thanks to your suggestion, we have added this sentence to the manuscript. (See details for line108-109)

(10) Line 120 TBLASTX requires a nucleotide query sequence and a nucleotide database, not a protein database. Perhaps you mean non-reduntant nucleotide database at NCBI (nr/nt)? Or was it a different kind of BLAST?

Reply: We used BLASTX for the blast in NCBI. This method involves translated nucleic acids to proteins. For this reason, we changed “TBLASTX” to “BLASTX”. (See details for line 119)

(11) Lines 127 – 128 a verb is missing in this sentence

Reply: Thanks for the heads up, we've revised the sentence:

A computer simulation model was generated using the I - TASSER online service. ( See details for line 127-128)

(12) Lines 161 – 171 Thank you for clarifying in your comments the determination of primer efficiency. I have no doubt you have done a good job, however this needs to be reported, even briefly, for the purpose of scientific transparency and replicability, in case other researchers wish to use your primers and assays. In this section a line could be inserted/added, such as:

‘The amplification efficiencies of of primers were greater than 90% (Cytb 95.6%; SjCyt 103.0%; SjCL 101.9%). The expected PCR product was in the range of 200-300 bp. All reactions were conducted in triplicates.’

Reply: Thank you for your suggestions, which we have added to the manuscript. (See details for line 169-172)

(13) Line 211 transformed

Reply: Thanks for the heads up, we have made modifications. (See details for line 214)

(14) Line 212 What does it mean by chemical method? Which chemical? Please specify.

Reply: I'm very sorry, it was my mistake to express that we used the chemical: ampicillin (Amp) antibiotic for positive screening after transformed. However, our transformed method is "Heat shock", which is the most commonly used transformation method, so we changed it in the manuscript. (See detail for line 215)

(15) Line 236 - 240 Statistical analyses

When you mention the three groups here the way they are now, it is confusing. No need to explain experimental design here, as it was already done in materials and methods. Also, all that you have done with the data needs to be reported here. Thank you for performing the log2 transformation on qPCR time course data, but it needs to be performed for all qPCR data. Even western blot – as it was standardized to beta actin? Actually, not sure, it depends on normality tests.

Reading in retrospect my comment, I see how it might have been misunderstood to lead you to do it only for part of the data. All that you have analyzed with ANOVA, first you need to check normality and equal distribution of variances – ALL experimental data and then apply a transformation if necessary, and it needs to be mentioned in this section.

Also, the authors did not address the problem of multiple testing corrections. You have made many comparisons and the p value obtained must be corrected for this. Bonferroni or Benjamini-Hochberg corrections are usually used. Please do so and mention this in the statistical analyses section.

So this section could be something like:

The statistical analyses of the data was performed by using one-way ANOVA in GraphPad Prism 9.5 (Graphpad software Inc., San Diego,CA, USA). The normal distribution and variance chi-square tests using SPASS were performed to ensure the data met ANOVA assumptions. The qPCR data were log2 transformed. The data were represented as mean±standard deviation, and the p < 0.05 was considered as statistically significant after correction ….?

Reply: Thank you for providing us with some valuable advice! We have provided a point-by-point response to your comments:

  1. We replaced Statistical analysis with Statistical analyses. (See detail for line 239)
  2. For the description problem you mentioned in the experimental design, we make the following changes: Randomly selected 5 living sea cucumbers. Different tissues, including dorsal epidermis, tube feet, intestine, respiratory tree, longitudinal muscle and coelomocytes, were collected from these sea cucumbers. All samples were preserved in RNAstore (TianGen Biotech, China) at −80°C for RNA extraction. Randomly selected 21 living sea cucumbers were taken out of seawater and put on a clean stainless−steel tray at ambient temperature (close to the marine water). Samples were collected from each group at distinct time intervals (0, 0.5, 1, 3, 5, 7 and 9 h). The dorsal epidermis and tube feet of sea cucumbers at each time point were collected and stored in RNAstore until RNA extraction. Conduct each experiment three times, and each repetition should be an independent trial. (See details for method 2.1)
  3. We also tested the western blot data for normal distribution and homogeneity of variance test, and did not do a log2 transformation since the data conformed to both.
  4. I'm very sorry for not understanding you before this, we took the RT-qPCR data (both tissue distribution and temporal expression), log2 transformed it, and did a normal distribution and homogeneity of variance test, and the data was conformed to both. (See details for Figure 7)
  5. Regarding p-value correction for multiple comparisons, the method we have chosen is Bonferroni, which is described in the manuscript as you suggested. (See details for 240-245)

(16) Line 293 Thank you providing a much better image. It is clear now.

The designation of colors for specific cystatin families in the figure is wrong, is it not blue for kininogen? Please check. Also, cystatins

Reply: Thank you for the reminder, it was my mistake and have changed it in the manuscript:

Green for family 1 (stefins); pink and yellow for family 2 (cystains); blue for family 3 (kininogens). (See the details for line 298)

(17) Line 379 So here perhaps you could reference one of those assays were it was proven cytb was a suitable housekeeping gene?

Reply: Thanks for your suggestion, we have added the corresponding reference to the manuscript. (See detail for line 382)

(18) Line 385 The legend is missing in Figure C and D.

Please do not forget to log2 transform data for A and B as well.

Reply: Thanks for your reminder, we have added graphical legend in Figure C and D, as well as log2 transformed the data of Figure A and B and re-analyzed the statistics.

(19) Line 394 Here it states the temporal data was normalized against 0 hours, but it is not very clear. Please expand the explanation. You could also add this in the figure caption.

Reply: Thank you for your suggestion, we have explained the normalization method in the manuscript:

Cytochrome b was used as a standard, and normalized to the expression level of 0 h by the ratio method. (See details for line 400-401).

(20) Line 454 Figure 9. Thank you for improving this image. Please add in figure caption what does the arrow mean. Also designate some other features that might be important.

It is still not exactly clear what Figure 9C shows, please explain how was these quantitative analyses of the immunofluorescence performed. What are the pluses and the minuses, etc. It could fit in the figure caption.

Reply: Thanks for your suggestion, we have added to the materials and methods of the manuscript (2.7): The number of fluorescent dots was counted using the software ImageJ and normalized by the number of nucleus. (See details for line 210-211)

In addition, we have added explanations to the caption of Figure 9:

Where arrows indicate peri−nucleus and circles indicate distal−nucleus. (See details for line 464-465)

(21) Line 462 please do mention here that: The purified rSjCyt protein was confirmed by SDS-PAGE, and western blot validation was conducted (data not shown).

Reply: Thank you for your suggestion, which we have added to the manuscript (See details for 472-473)

Thank you for improving the discussion. Please just check the grammar after track changes are removed, I think there are few that remained.